# Structural insights into cauliflower mitoribosome in translation state and in association with a late assembly factor

Vasileios Skaltsogiannis [1], Philippe Wolff[2], Tan-Trung Nguyen [3], Nicolas Corre[1], David Pflieger [1], Todd Blevins [1], Yaser Hashem [3], Philippe Giegé [1] ✉ & Florent Waltz [4] ✉

Ribosomes are key molecular machines that translate mRNA into proteins. Mitoribosomes are specific ribosomes found in mitochondria, which have been shown to be remarkably diverse across eukaryotic lineages. In plants, they possess unique features, including additional rRNA domains stabilized by plant-specific proteins. However, the structural specificities of plant mitoribosomes in translation state remained unknown. We used cryo-electron microscopy to provide a high-resolution structural characterization of the cauliflower mitoribosome, in translating and maturation states. The structure reveals the mitoribosome bound with a tRNA in the peptidyl site, along with a segment of mRNA and a nascent polypeptide. Moreover, using structural data, nanopore sequencing and mass spectrometry, we identify a set of 19 ribosomal RNA modifications. Additionally, we observe a late assembly intermediate of the small ribosomal subunit, in complex with the RsgA assembly factor. This reveals how a plant-specific extension of RsgA blocks the mRNA channel to prevent premature mRNA association before complete small subunit maturation. Our findings elucidate key aspects of translation in angiosperm plant mitochondria, revealing its distinct features compared to other eukaryotic lineages.

Mitochondria are vital organelles found in nearly all eukaryotic organisms, where they play a central role in energy regeneration and metabolism[1,2]. Through oxidative phosphorylation, these organelles are responsible for recycling ADP to ATP, the energy currency of the cell[3]. They are also involved in numerous core cellular functions, such as calcium signaling, apoptosis, the synthesis of heme and other essential metabolites[4]. In photosynthetic eukaryotes, such as plants and algae, the role of mitochondria is closely linked to that of chloroplasts[5,6]. While chloroplasts are responsible for producing energy-rich molecules, they are not capable of carrying out all the necessary metabolic processes on their own[7]. To overcome this limitation, photosynthetic organisms rely on their mitochondria to carry out these metabolic processes[8].

Due to their endosymbiotic origin, mitochondria possess their own genomes and specific gene expression machineries[9]. Among these, the mitochondrial ribosome (called mitoribosome hereafter) is an essential molecular machine that synthesizes proteins within the organelle. In particular, some genes encoding subunits of respiratory complexes are encoded in mitochondrial genomes, and mitoribosomes are thus essential for the biogenesis of oxidative

[1]Institut de biologie de moléculaire des plantes CNRS, Université de Strasbourg, 12 rue du général Zimmer, Strasbourg, France. [2]Architecture et Réactivité de l'ARN, Institut de Biologie Moléculaire et Cellulaire du CNRS, Université de Strasbourg, 2 allée Konrad Roentgen, Strasbourg, France. [3]Institut Européen de Chimie et Biologie, Université de Bordeaux, 2 rue R. Escarpit, Pessac, France. [4]Biozentrum, University of Basel, Spitalstrasse 41, Basel, Switzerland. ✉e-mail: philippe.giege@ibmp-cnrs.unistra.fr; florent.waltz@unibas.ch

phosphorylation complexes[10,11]. Because respiratory proteins are often intrinsic membrane proteins, mitoribosomes have evolved and specialized for the synthesis of these hydrophobic proteins[12–15]. Recent advances in cryo-electron microscopy (cryo-EM) have enabled the study of mitoribosomes at high resolution, providing valuable insights into their composition, architecture and function. Mitoribosomes were characterized in a wide diversity of eukaryotes, i.e., in mammals, fungi, kinetoplasts, ciliates, angiosperm plants (i.e., flowering plants) and green alga[13,16–23]. These studies have revealed the remarkable diversity of mitoribosome composition and structures[24,25]. Still, a common feature of mitoribosomes is their high protein content, with at least 60 specific ribosomal proteins occurring in the respective mitoribosomes as compared to the bacterial ribosome[26]. Interestingly, helical repeat proteins such as pentatricopeptide repeat (PPR) proteins are prevalent among mitoribosome-specific proteins[9]. The initial description of mammalian mitoribosomes has shown that ribosomal RNA content is extremely reduced in these eukaryotes[16,17]. This led to the hypothesis that the contraction of rRNA in mitoribosomes could be compensated by increased numbers of ribosomal proteins. Likewise, this suggested that mitoribosomes could follow an evolutionary trend to reduce rRNA size, that could ultimately result in the complete loss of rRNA[27]. Even though this model applies to several eukaryotes, it was contradicted by the characterization of fungal and, in particular, of plant mitoribosomes that contain both increased numbers of proteins and expanded rRNAs[18,20,28].

Plant mitoribosomes were characterized at the biochemical and structural level in the closely related Brassicaceae species Arabidopsis and cauliflower[20,29]. This revealed a very distinctive, enlarged mitoribosome with 10 plant-specific ribosomal proteins, among which 8 are PPR proteins, although 4 of them could not be structurally assigned. The small subunit (SSU) is particularly singular as it is larger than the ribosomal large subunit (LSU) and contains a distinctive plant-specific elongated domain on the head of the SSU[30]. Despite these advances, many features of plant mitochondrial translation remain elusive. For instance, how these ribosomes assemble, are tethered to the membrane, or the way mRNAs are recruited to the SSU remain speculative. Likewise, the molecular details of the actual translation process catalyzed by the plant mitoribosome are unknown.

In this work, we attempt to address some of these open questions. We produce a sub-3 Å cryo-EM structure of the cauliflower mitoribosome, which reveals details about rRNA modifications and r-proteins, particularly 4 distinct plant-specific PPR proteins. Along with the improved resolution, we also describe the structure of a stalled mitoribosome with mRNA and a P-site tRNA, in the presence of chloramphenicol. Moreover, a very late assembly intermediate of the SSU, bound with the assembly factor RsgA, was obtained, revealing the function of its plant-specific domain and the mode of action of this factor in the context of a mitoribosome.

## Results and discussion

### Strategies for ribosome stalling and structure determination

To improve our understanding of plant-specific translation processes, we purified mitochondria from cauliflower and subsequently purified mitoribosomes in the presence of chloramphenicol or non-hydrolysable GTP analog to stall translation and/or other processes that might require GTP hydrolysis. Four datasets were collected, datasets 1 and 2 with untreated sample, 3 with chloramphenicol treated sample and 4 with GTP analog treated sample (see methods), resulting in the three main structures presented here: a full ribosome structure without tRNA resolved to 2.1 Å obtained with sets 1 and 2 (Fig. 1a, b, Supplementary Figs. 1, 2), a stalled ribosome harboring a P-site tRNA and mRNA resolved to 3.0 Å, obtained with set 3 (Supplementary Fig. 3), and a late SSU assembly intermediate in presence of the assembly factor RsgA obtained at 2.2 Å with set 4 (Supplementary Fig. 4).

Given the large flexible extensions of this mitoribosome, we first used 3D Flexible Refinement[31] (Supplementary Movie 1), to investigate the major modes of motion of its different domains. On top of the classical ratcheting motion, we observed the head extension motion, but also the movement of the SSU foot and of the LSU back extension. Then, extensive masking and focused refinements revealed parts of the mitoribosome that remained poorly resolved in the previous studies, notably the SSU head extension. Given the high local resolution for the different parts of the ribosome (up to 1.8 Å locally, permitting to place ions and solvent molecules (Fig. 1c, Supplementary Figs. 1, 2)), we adopted an unbiased approach to rebuild and identify ribosomal proteins. ModelAngelo[32] was used to sequence the proteins directly from the density, and the resulting models were verified against the protein candidates detected by MS[29]. This identified a total of 85 proteins (47 in the LSU and 38 in the SSU), which we consider to be the complete set of core r-proteins of this mitoribosome (complete table of proteins Supplementary Tables 2, 3).

### Structural characterization of plant-specific ribosomal proteins

One of the most remarkable features of plant mitoribosome is the occurrence of many PPR proteins as core ribosomal proteins. This is in accordance with their high number in plants as compared to other eukaryotes, with hundreds of PPR in plants as compared to only tens in other taxa[33,34]. A major drawback of previous studies was that half of these ribosomal PPRs could not be visualized at sufficient resolution to structurally assign them unambiguously. This has been resolved here. It now allows us to fully apprehend the composition of the plant-specific domains of mitoribosomes. For instance, in the large subunit, the remodeled domain III of the 26S rRNA on the LSU back is highly flexible (see Supplementary Movie 1) and was thus poorly resolved. There, the rRNA is remodeled mainly by PPR proteins mL104 (rPPR9) and mL101 (rPPR4). Using focused refinement, we resolved it in greater detail, notably their interaction with the rRNA. Beyond these PPR proteins that are specific to plants, it also revealed the presence of a herein identified r-protein (Fig. 1d, Supplementary Fig. 5a–d). This protein is an isoform of bL25m, that we name bL25-2m, interacting with both the remodeled domain III and mL101. There, it shapes and stabilizes the remodeled domain III by mimicking the interaction of bL25m with the 5S and 26S rRNA located between the L7/L12 stalk and the base of the Central Protuberance (CP) (Supplementary Fig. 5d). Focused refinement on the L7/L12 stalk revealed the presence of mL54 (conserved with algae, mammals and yeast) which was previously identified only by mass spectrometry (Supplementary Fig. 5e). The largest PPR protein of the LSU, mL102 (rPPR5), contacts the 26S rRNA at different positions of the LSU, mostly through its N-ter part, revealing how it shapes domain I plant-specific helices p15, p17 and p21-22 (Supplementary Fig. 6a). uL2m is split into two parts in angiosperm plants, one encoded in the mitochondrial genome (N-terminal portion) and one in the nuclear genome (C-terminal portion) and forms an intricate network of protein-protein interactions with multiple r-proteins. First, it interacts with mL87 and then extends far away from the uL2 region to closely interact with bL28m, bL9m, as well as mL102 (Supplementary Fig. 6c).

Likewise, the SSU model was considerably improved with the identification of several proteins that were previously modeled but not identified, i.e., for the SSU head extension (Fig. 1e). In angiosperms, the 18S rRNA harbors a large extension at the tip of the SSU's head, rooting from h39 of domain 3'M. Here, we resolve this specific domain to approximate 4 Å (compared to the previously reported 8 Å). This allowed the unambiguous identification of the two r-proteins located there (rPPR6 and rPPR8) (Supplementary Fig. 7a). mS80 (rPPR6) is positioned at the base of the head extension, whereas mS81 (rPPR8) wraps around and follows the shape of the head extension. Two PPR modules of the latter are found (shown in orange in Fig. 1e), the longest one (aa 219–656) follows the two parallel RNA helices, while the second

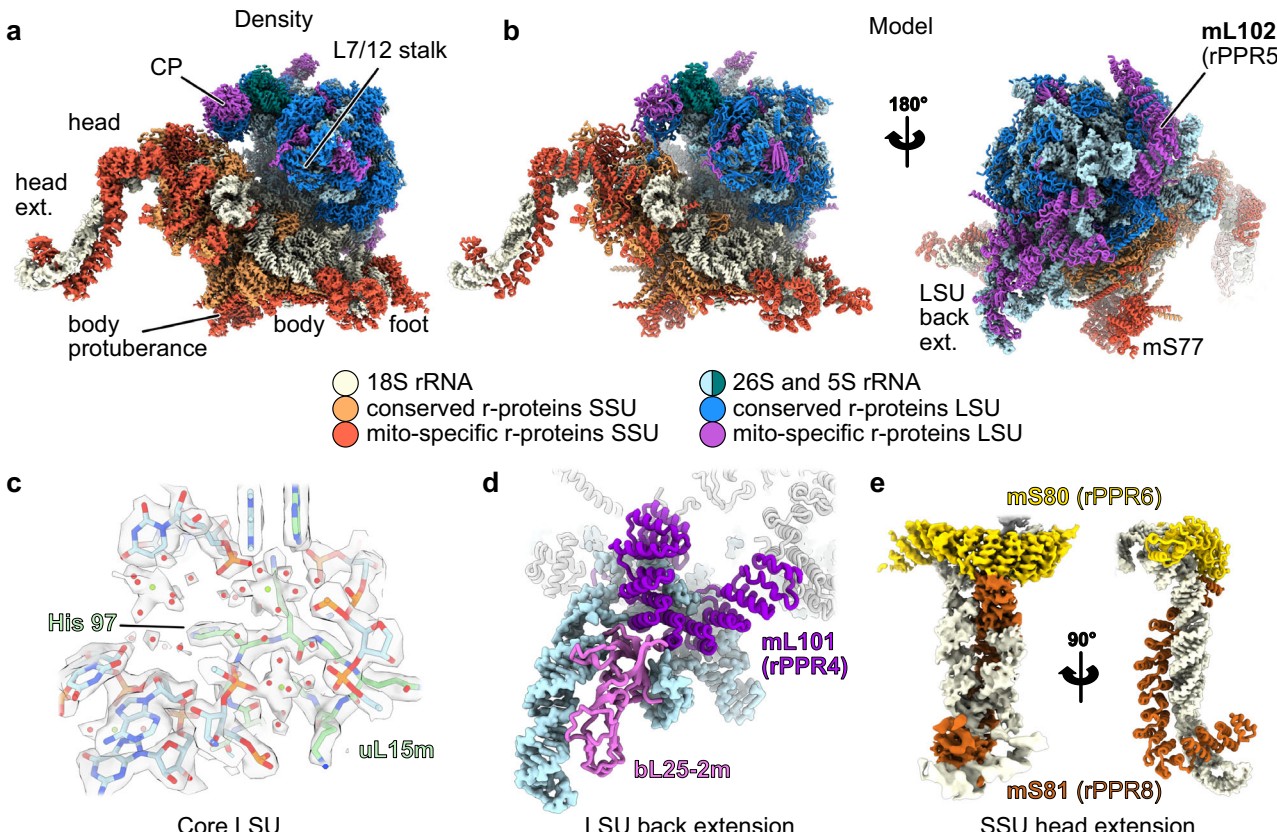

**Fig. 1 | High resolution structure of the plant mitoribosome. a** Composite high-resolution map of the cauliflower mitoribosome with components of the SSU shown in beige, coral and red and components of the LSU shown in blue shades and purple. **b** Resulting atomic model with proteins shown in cartoon representation and rRNA shown as surface. **c** Close-up view of the core of the LSU, showing solvent molecules and ions. Red spheres represent water, while green spheres represent magnesium ions. **d** View of the LSU back extension, with the remodeled rRNA Domain III and the identified r-protein bL25-2m. **e** View of the SSU head extension, on the left as a density, and on the right a rotated view as an atomic model.

one (aa 76–218) is bent almost 90° relative to the first one, and helps form the tip of the SSU head extension. This plant-specific domain of unknown function might serve as a hub for the recruitment of trans-factors, possibly involved in translation-related processes or other RNA-related processes.

The SSU foot also harbors plant-specific features. We could identify that it involves mS76 (rPPR1) and mS83 (rPPR10), which were previously modeled as Unknown proteins (Supplementary Fig. 7b). mS76 contributes to the stabilization of the h6 helix of domain 5′ of the 18S rRNA whereas mS83 mostly contributes to the stabilization of h44 tip, as well as h6. A recent study has performed the functional analysis of mS83 (called mTRAN by the authors)[35]. A global impairment of mitochondrial translation in mS83 mutants was observed, and mS83 was found to be able to bind RNA in vitro. The authors proposed that mS83 is a general factor for translation initiation in plant mitochondria. However, here we place mS83 at the tip of the foot of the SSU, interacting with plant-specific rRNA expansion segments, which is not consistent with a role in translation initiation.

Most remarkably, we could identify that the mRNA channel exit harbors mS77, another plant-specific PPR protein (Supplementary Fig. 7c). It wraps around the SSU body, rooting from the SSU protuberance, where it is anchored and extends in the back of the SSU, close to the mRNA exit channel, where its PPR domain is located. Originally listed as two proteins in UNIPROT (Q8LEZ4, NFD5, N-ter part, making part of the body protuberance and Q940Z1, PPR51, C-terminal part containing the PPR motifs), we instead identify it as a single protein as annotated in the UNIPROT entry A0A654ECT9. Moreover, at low threshold we also observe an additional density at the tip of the PPR

domain, suggesting that this protein might act as a platform to recruit specific plant translation factors, or allow polysome formation (Supplementary Fig. 7c). Next to this protein, we also identify a small portion of the N-terminal part of mS86 (GRBP6), (residues 1 to 28, the protein being 155aa), sandwiched in between uS2m and mS23 (Supplementary Fig. 7c). mS86 was previously identified through mass spectrometry but not resolved in the structure. Similar to mS77, its position next to the mRNA exit channel may suggest an interaction with mRNA. These additional plant-specific proteins might also have been recruited to the exit channel to compensate for the absence of uS1.

Additionally, on the head of the SSU, we fully resolve the composition of the protein protuberance located close to the beak (Supplementary Fig. 8a, b). It is formed by mS31/mS46, that is also herein identified, mS35, and uS3m, which was predicted to form part of this protuberance (Supplementary Fig. 8b). Together, they form an intricate network of proteins close to the mRNA entrance channel. Finally, the improved resolution reveals the contacts between the LSU's CP to SSU head via bL31m, mL40 and mL46, and unravels the presence of ATP in mS29 (Supplementary Fig 8a), as also found in human mitoribosomes[36].

Altogether, out of the 85 mitoribosomal proteins, 53 are conserved with bacteria, 22 are shared with the mitoribosome of other eukaryotic species, and 10 proteins are specific to plants, among which 8 are PPR proteins (Supplementary Table 2 and 3). It is remarkable that 7 of these plant-specific PPR proteins stabilize rRNA expansion segments also specific to plants, thus creating distinct plant-specific domains. This suggests a co-evolution process between the expansion

of rRNAs and the recruitment of PPR proteins as ribosomal proteins in plant mitochondria. Only a single PPR protein, mS77, does not bind rRNA. Structural data might point out a role for the recruitment of mRNA and/or protein trans factors, possibly for a plant-specific translation initiation process.

## Cauliflower mitoribosome contains a set of 19 rRNAs modifications

rRNA modifications are hallmarks of subunit maturation towards a translationally active ribosome[37,38]. To date, the rRNA modifications present in plant mitochondria have not been described. Our cryo-EM map resolution allowed us to identify in total 18 rRNA modifications, 6 in SSU and 12 in LSU (Fig. 2a, b). To confirm the modifications detected by cryo-EM and/or identify additional ones, we used nanopore direct RNA sequencing of the native cauliflower rRNAs. Nanopore RNA sequencing has been successfully applied before to detect RNA modifications, for instance, in ribosomes[39–42]. This second approach indicated 28 positions with putative modifications, 8 in SSU and 20 in LSU (Fig. 2c, Supplementary Figs. 9, 10). Finally, a third approach using mass spectrometry was used to detect rRNA modifications. Results identified 20 rRNA modifications, 7 in SSU and 13 in LSU (Fig. 2c, Supplementary Fig. 11). Discrepancies between the different approaches can be due to insufficient local resolution in some parts of the structural model, by the fact that some sequence contexts associated with some modifications occasionally do not result in detectable identification of modifications by nanopore sequencing[41,43] and by the inability to detect modifications by mass spectrometry in some sequence contexts. The aforementioned approaches were employed to detect the position and the type of each modification. When the latter was not adequately supported by our experimental methods, the type of modification was assigned according to its equivalent residue of the *E. coli* ribosome, for the conserved modifications (Supplementary Table 4), or left unassigned for the plant-specific ones. Cryo-EM densities for the position identified by MS or nanopore but not by cryo-EM are shown in Supplementary Fig. 12.

In total, structure inspection, nanopore sequencing and mass spectrometry resulted in the identification of 39 positions with putative rRNA modifications, as presented in Fig. 2, Supplementary Fig. 12 and Supplementary Table 4. Of these, 19 modifications were robustly supported by at least two approaches and considered as consensus modifications. Surprisingly, this number deviates considerably from the ones of the scarcely modified mammalian or yeast mitoribosomes that contain 10 and 5 rRNA modifications, respectively[40,44–47].

We found 9 types of base modifications, including a network of 8 pseudouridines (Ψ) and 3 types of 2′-O ribose methylation (Nm). In the structural analysis, Ψ are identified through the presence of a water molecule, close to a nitrogen atom located at position 3 of the base after uridine isomerization, while in the nanopore analysis, Ψ are mostly read as cytidines[48] and their detection by mass spectrometry is allowed by specific cyanoethylation of Ψ that changes the mass of the respective RNA fragments. Like their bacterial and cytosolic counterparts, rRNA modifications of the plant mitoribosome form clusters close to functionally important sites (Fig. 2b). Near the decoding center, we located the universally conserved methylated residues $m^6_2A1827$ and $m^6_2A1828$. These adjacent modified adenosines aid the P-site in acquiring its mature conformation by establishing contact between helices h44 and h45. In this vicinity, $m^2G1825$ (although not in the consensus, it is clearly identified by cryo-EM) of h45 and two modifications located at the base of h44, $m^3U1807$ and $m^4Cm1664$, were further identified and characterized in the context of a mitoribosome (Fig. 3). $m^4Cm1664$ interacts directly with the mRNA and has been proposed to fine-tune the decoding center in bacteria by preventing translation initiation at the alternative initiation codon AUU and by maintaining the reading frame[49]. Moreover, mass spectrometry indicated the presence of a methyl group at the position C1671.

Likewise, nanopore sequencing revealed substantial base calling errors, although below our 10% threshold, for the flanking positions A1670 and A1672 (ΔCallError: 9.1% and 9.9%, respectively), pointing to the presence of a modified base in the vicinity (Supplementary Fig. 10a). The plant-specific (m)C761 is likely to participate in the formation of intersubunit bridges, upon monosome assembly.

Another cluster of rRNA modifications was observed around the peptidyl transferase center (PTC). Among them, we were able to identify the highly conserved Gm2560 and Um2857 at helices 80 and h92, respectively. Gm2560 is in the P-loop region and supports the positioning of the P-tRNA by interacting with its CCA-end, whereas Um2857 resides in the A-loop region and assists in the accommodation of the tRNA at the A-site. Interestingly, mammalian mitoribosomes possess an additional 2′-O methylation at the adjacent G2858 residue, which we could not model due to lower resolution in that region. Nonetheless, our nanopore data point to the existence of a modification on this specific residue; thus, we do not rule out the existence of Gm2858. The A-tRNA accommodation is further stabilized by the pseudouridine Ψ2885, similar to human mitoribosomes[47]. The domain IV of the LSU rRNA establishes the intersubunit bridges of group B2, crucial for ribosome assembly and translation. In this region, we identified the methylated sites $m^5U2257$ (h70) and the pseudouridine Ψ2188 (h68), which is plant-specific. Moreover, our nanopore and mass spectrometry data show the presence of four additional pseudouridylation sites in helix 69. These could not be observed by cryo-EM, as the resolution is lower in that area of the map, but were confirmed by another study[50]. Finally, the presence of the modified nucleotide $m^2A2808$ was observed at the PTC (Fig. 3). The methyl group of $m^2A2808$ extends its stacking range, allowing the interaction with A2377 (Supplementary Fig. 11), thus promoting the folding of their cognate single-stranded rRNA segments to form the peptide exit tunnel wall.

Overall, the plant mitoribosome is highly modified and has retained a significant number of the modifications present in its bacterial ancestor. Thus, it appears to have followed an evolutionary route that allowed for the preservation of many bacterial modifications in direct contrast to its mammalian and yeast counterparts, who were able to retain only a few. This also suggests that many bacterial modification enzymes must be conserved in plants (putative enzymes based on homology are presented in Supplementary Table 4), or have been replaced by other ones, to maintain these modifications.

## Identification of plant-specific features of the translating mitoribosome

To trap ribosomes in translation states, we collected a dataset for mitoribosomes purified from mitochondria and ribosomes treated with chloramphenicol, a classical inhibitor of bacterial-type ribosomes. Chloramphenicol binds to the PTC of the ribosome, stalling translation[51]. During data processing, we identified a subclass corresponding to mitoribosomes in a P-site tRNA-accommodated state, which we resolved to approximately 3 Å resolution (Fig. 3a, Supplementary Fig. 3). This allowed us to build a detailed model that includes the tRNA, the messenger RNA path, the binding site of chloramphenicol, and the peptide route through the peptide channel.

At the PTC, within the LSU, we fully resolved the binding site of chloramphenicol, responsible for stalling the ribosome in the P-site tRNA-accommodated state (Fig. 3b). At this site, the first four amino acids of the nascent peptide chain were clearly visible. Since this represents a mix of averaged densities from different nascent proteins, we modeled it as poly-alanine. Since mitoribosomes were treated with chloramphenicol, we were able to resolve the antibiotic's binding site, revealing that its mode of action is nearly identical to its binding in classical bacterial ribosomes. This is not surprising, as the PTC is one of the most highly conserved regions of the ribosome, alongside the decoding center[52]. The chloramphenicol binding in the mitoribosome

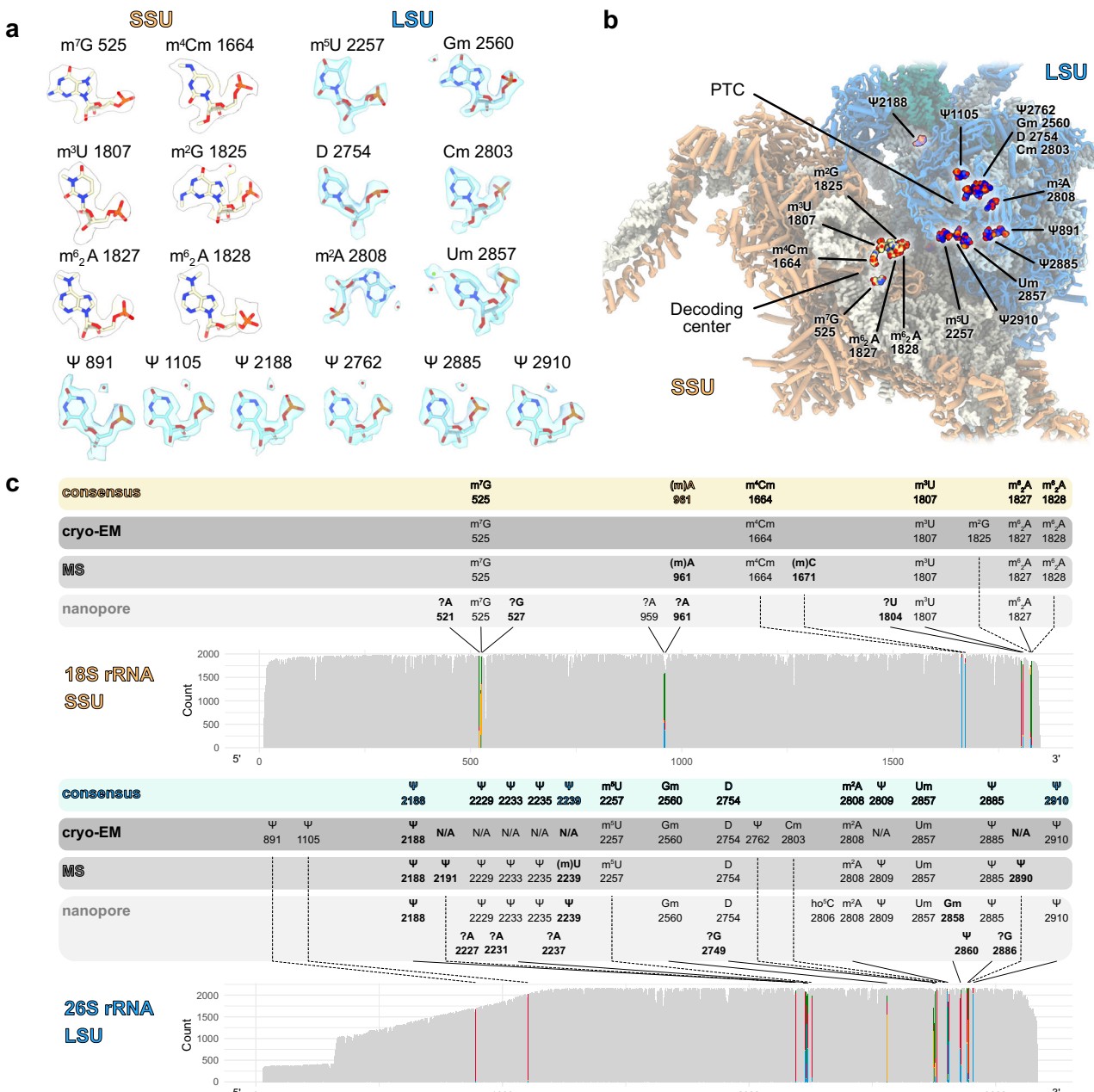

**Fig. 2 | Plant mitoribosome rRNA modifications. a** All the rRNA modifications identified from the cryo-EM map are shown in their respective densities with modifications in the LSU 26S rRNA shown in light blue and in the SSU 18S rRNA shown in beige. **b** Positions of the identified modifications are indicated on the structure of the mitoribosome. **c** Modifications identified by nanopore, cryo-EM and mass spectrometry. Plots represent the base composition of native rRNA sequences compared with in vitro rRNA transcripts (IVT) sequences obtained by nanopore direct RNA sequencing. The x-axis shows nucleotide positions along the rRNA; the y-axis shows the per-base read depth (reads overlapping each position). All positions of interest−i.e., indicated as modified by any of the detection methods −are presented as colored percentage stacked bars in the otherwise gray read depth chart. Color categories in such highlighted bars correspond to four possible base calls obtained at the rRNA position: Green, A; Red, U; Orange, G; and/or Blue, C. Non-reference base calls are positions of interest identified by nanopore direct

RNA sequencing that may indicate base modifications. The nature and positions of modifications observed in the structure are indicated in black, while modifications observed by nanopore are indicated in gray. "?" corresponds to putative RNA modifications that could not be characterized. "N/A" represents positions that could not be visualized by cryo-EM because of insufficient local resolution. For the MS analysis, positional isomers of methylated nucleotides were indicated according to equivalent bacterial rRNA modifications. For nanopore DRS analysis, the type of modification (other than Ψ) was assigned according to equivalent bacterial rRNA modifications. Modifications that were observed with at least two approaches are shown in the consensus bar. The 9 modifications that are specific to the plant mitoribosome are indicated in bold. Details for each position identified by nanopore sequencing are shown in Supplementary Figs. 9, 10 and listed in Supplementary Table 4.

is positioned similarly to its location in bacterial ribosomes, reinforcing the conserved nature of its inhibitory mechanism across species.

In the decoding center of the small subunit (SSU), we also captured the mRNA being translated, from which we resolved six nucleotides, three of which were interacting with the anticodon loop

of the stalled tRNA (Fig. 3c). Similar to the peptide observed at the PTC, these nucleotides represent an averaged mix of densities from different mRNAs. Nonetheless, we were able to observe how they are accommodated within the small subunit. Here, conserved residues of the 18S rRNA, G956, and C1746 help stack and position the tRNA. The

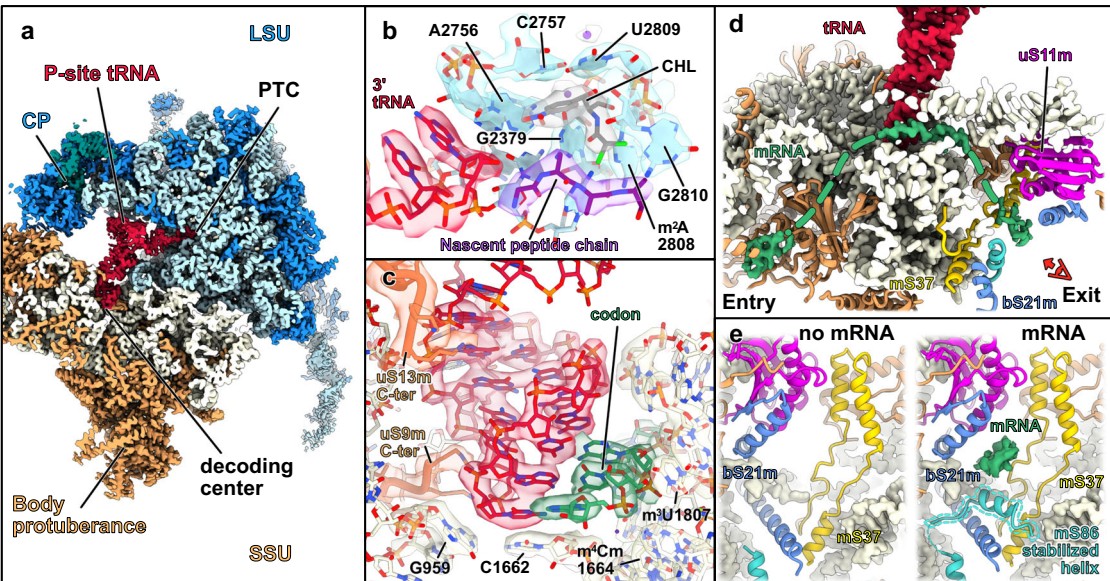

**Fig. 3 | Plant mitoribosome in a translating state. a** Composite map of the mitoribosome with the SSU shown in beige, the LSU in blue and a tRNA in the P-site shown in red. **b** Close-up view of the PTC with bound chloramphenicol (CHL) shown in gray and the stalled nascent peptide chain in purple, shown in their density. The nucleotides forming the chloramphenicol binding pocket are annotated and shown in their density. Potassium ions are shown as purple balls. **c** Close-up view of the decoding center highlighting the interaction between the codon in green and the anticodon in red as well as uS13m and uS9m probing the anticodon loop, shown in their density. **d** Cross-section view of the SSU highlighting the mRNA channel, with resolved mRNA residues depicted in green at the decoding center, as well as at the channel's entry and exit points. **e** Close-up views of the mRNA channel exit, viewed from the red eye in (**d**), comparing the structures with and without mRNA. In the presence of mRNA, the mS86 helix is stabilized.

ribosomal protein uS13m probes the base of the tRNA anticodon region with its C-terminal extension, behaving in a manner identical to bacterial ribosomes, while uS9m, which has a slightly extended C-terminus compared to its bacterial counterpart, probes a different region of the tRNA anticodon (Fig. 3c).

In contrast with the PTC and the decoding center, which are globally conserved as compared to bacteria, the mRNA channel revealed plant-specific features for the accommodation of mRNA. At a lower threshold, we could visualize densities corresponding to mRNA at the entry and exit sites of the mRNA channel, allowing us to trace the full putative mRNA path within the SSU (Fig. 3d). At the entry point, the conserved core of proteins uS4m, uS5m, and uS3m (located on the head of the SSU) form most of the entry channel. Extensions of uS5m and a portion of uS4m, which form parts of the SSU body protuberance, further delineate and extend the entrance of the mRNA channel. On the SSU head, uS3m, together with mS31/46 and mS35, contributes to the SSU head protuberance near the beak. Notably, 300 amino acids from the N-terminus of mS31/46 remain unresolved and point toward the solvent, near the mRNA entrance (Supplementary Fig. 8b). Altogether, the SSU body and head protuberances located near the mRNA entry channel could play a role in species-specific recruitment of mRNA or polysome formation, which has been observed in situ in Chlamydomonas[53]. On the opposite side, at the mRNA exit, the ribosomal proteins uS11m, bS21m, and the mitochondria-specific mS37 form the exit channel. Additionally, we observed a larger portion of mS86 in the ribosome charged with tRNA and mRNA compared to empty ribosomes (Fig. 3e). An additional helix in mS86 is stabilized up to residue 54 (compared to 28 amino acids in the empty ribosome) at the mRNA exit channel. The fact that this protein is stabilized in the stalled ribosomes, where mRNA is present, supports the hypothesis that mS86 may be important for mRNA interaction. These plant-specific features of the mRNA channel, together with the occurrence of mS77 at the exit of the channel, likely point out specific processes, i.e., for the recruitment of mRNAs to plant mitoribosomes.

## Identification of a plant-specific process to prevent mRNA association before completing small subunit maturation

Mitoribosome biogenesis is coordinated by the orchestrated action of a series of assembly and auxiliary factors that guide the rRNA to adopt its mature conformation and, in parallel, drive the integration of the complete set of ribosomal proteins to generate two functional ribosomal subunits. Structural and mechanistic insights into the assembly pathway of mitoribosomes have been recently provided in kinetoplastid[22,54–57], human[37,38,58–65] and yeast[37,66]. Nevertheless, what assembly factors participate in the biogenesis of plant mitoribosomes and the mechanisms under which they operate remain elusive. To address these questions, we tried to capture native assembly intermediates by stalling the assembly process of the plant mitoribosome. We targeted assembly factors of the GTPase family, known to be important enzymes for ribosome maturation[67]. To do so, we isolated mitoribosomes in the presence of a non-hydrolysable GTP analog, GTPγS. Even though no clear new state of the fully assembled mitoribosome could be identified, analysis of the GTP analog SSU-only fraction revealed a subclass of 208,855 particles with an additional density located near the decoding center (Fig. 4a–g). This class reached 2.2 Å resolution globally, allowing us to directly identify the additional factor from the density as the assembly factor RsgA (Fig. 4b). RsgA adopts a conformation where its central GTPase domain docks at the decoding center of the SSU (Fig. 4g) while its oligonucleotide/oligosaccharide binding-fold domain (OB-domain) spans towards the uS12m (Fig. 4d) and the Zinc-binding domain interacts with the head of the SSU via helix 29 (Fig. 4e). This conformation is similar to the one observed in bacterial ribosomes bound to RsgA[68–70]. The bacterial RsgA is considered to act as a late checkpoint for the maturation state of the decoding center, via its direct interaction with h44[68]. Consistent with this notion, we observed a similar interaction of Arg[84] with A1802 of h44. Also, considering the overall maturation state of the SSU, the plant RsgA acts upon the late steps of the SSU biogenesis. Despite the similarities shared by the two homologs, plant RsgA possesses a 66 amino-acid long C-terminal extension that

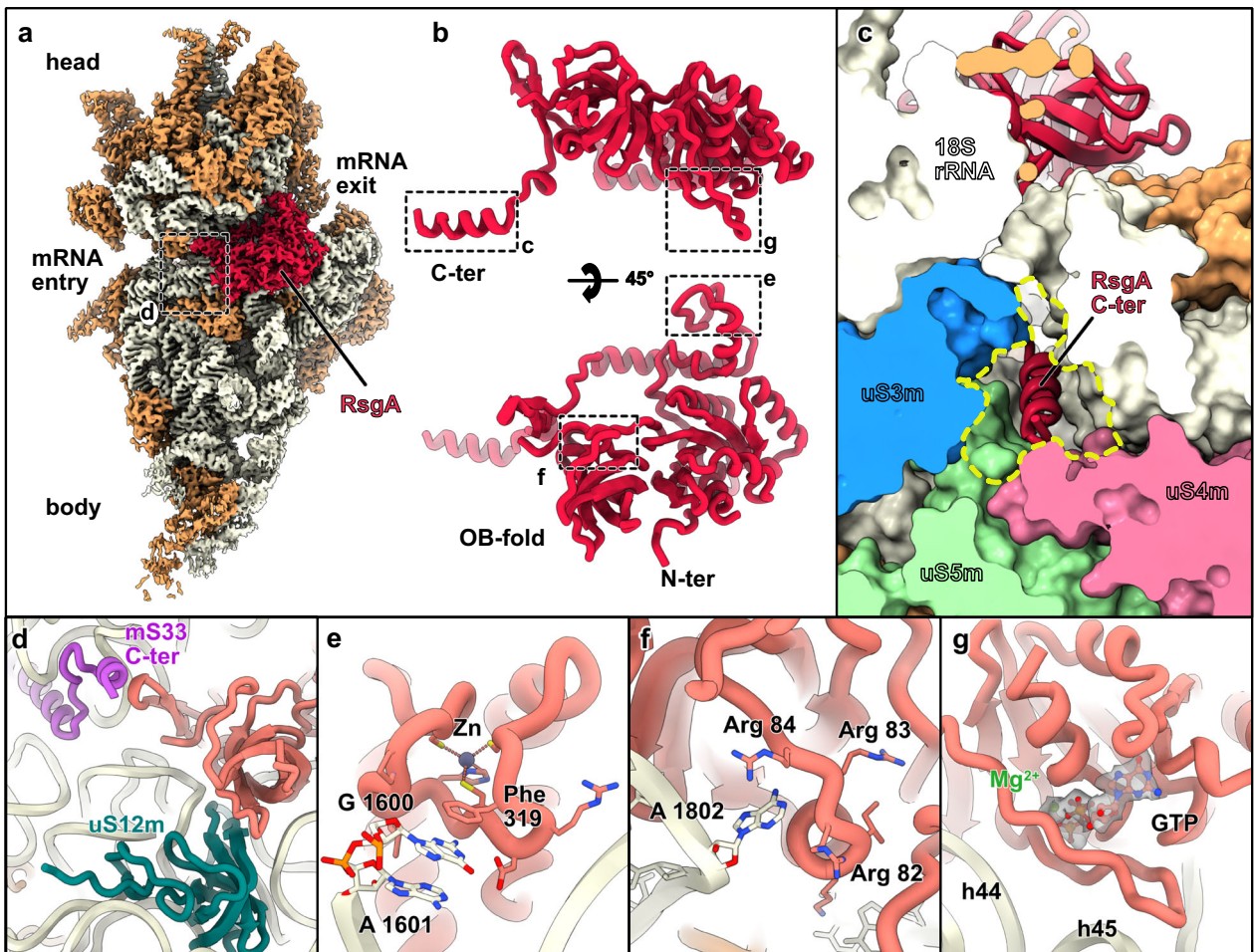

**Fig. 4 | Late stage SSU maturation complex in presence of RsgA. a** Composite map of the SSU, with r-proteins shown in coral, rRNA shown in beige, and RsgA shown in red. **b** Close-up view of the RsgA atomic model, shown from two different point of views. Dashed boxes indicate further detailed views. **c** View of the RsgA C-terminus blocking the mRNA channel entry. **d**–**g** Close-up views of the different positions probed by RsgA. **d** RsgA protein-protein contacts with mS33 C-terminus and uS12m. **e** Zinc-binding domain area, with Phe319 probing G 1600 on the head of the SSU. **f** Arg 84 from the OB-fold domain probing A1802 at the base of h44. **g** GTP binding site, with GTP analog shown in its density. In that region, RsgA probes the base of h44 and the tip of h45.

appears to have a dual function (Fig. 4c). First, it reinforces the positioning of RsgA by interacting with mS33 (Fig. 4d), and second, adopts a certain conformation that allows its entry into the mRNA channel. Remarkably, the plant-specific extension folds into an alpha-helix that threads into the mRNA channel via its placement in between uS3m, uS5m, and uS4m (Fig. 4c). Therefore, RsgA blocks the mRNA channel and prevents the premature accommodation of the mRNA during the assembly process. The function of sequestering the immature SSU particles from those that are fully functional is pivotal and is served by various mechanisms along the assembly pathway, mainly by making the mRNA channel inaccessible. In order to investigate the conservation of this specific process observed in cauliflower, we performed a phylogeny analysis of RsgA orthologs. While RsgA is highly conserved in bacteria, it is not found in eukaryotes such as yeast or human. It is in contrast conserved in the plant lineage. Moreover, the RsgA C-terminal extension appears to be widely conserved in land plants (Supplementary Fig. 13), although it seems to be absent from gymnosperms. Localization predictions suggest that RsgA might only be present in mitochondria from dicotyledon plants. In other plant groups, RsgA proteins are rather predicted to be plastidial, which suggests that a process similar to the one described here might occur in some plant groups for chlororibosome assembly. Interestingly, in humans, during the late stages of SSU biogenesis of the mitoribosome, the function held here by RsgA is performed by the assembly factor RBFA, which

has acquired a mitochondria-specific C-terminal extension that spans across the mRNA channel, preventing the mRNA recruitment. It is apparent that the plant RsgA and the human RBFA are two evolutionary unrelated assembly factors that have independently acquired lineage-specific extensions in their amino acid sequence. These extensions perform a similar function, expanding their basic functional repertoire in the assembly process. Thus, these assembly factors constitute a fine example of convergent evolution between plants and human mitochondria.

In this study, we provide insights into the function and assembly of the translation apparatus in an angiosperm plant using cryo-EM. While it was already known that plant mitoribosomes exhibit significant structural complexity, with expanded rRNA domains and numerous plant-specific ribosomal proteins, the data presented here provide significantly improved resolution and identify distinct r-proteins. Furthermore, we detected distinct features such as nucleotide modifications, revealing a high number of rRNA modifications in plant mitoribosomes as compared to mammalian and yeast mitoribosomes. These modifications are located near functionally critical regions such as the decoding center and the peptidyl transferase center, highlighting the evolutionary conservation of certain bacterial features in plant mitoribosomes. Although some modifications were acquired during the evolution of eukaryotes, most rRNA modifications represent ancestral features of mitoribosomes.

Furthermore, our study identifies the plant-specificities of key functional domains of the mitoribosome, the decoding center and mRNA path as well as the PTC, further elucidating unique aspects of translation in plant mitochondria. Finally, we captured a late-stage assembly intermediate of the mitoribosome, bound to the assembly factor RsgA. This interaction reveals the mechanism by which RsgA ensures the proper maturation of the small subunit by blocking the mRNA channel during assembly, thanks to a plant-specific domain.

Overall, we thus found that plant mitoribosomes diverged from bacterial ribosomes, from cytosolic ribosomes and from mitoribosomes in other eukaryote groups. As compared to bacteria, the plant mitoribosome PTC is globally conserved, thus reflecting its bacterial origin, but its mRNA channel is remodeled, i.e., with the presence of a plant-specific PPR protein at the exit of the channel. It also harbors additional plant mitochondria-specific domains that result from the expansion of rRNAs and the recruitment of PPR proteins to stabilize them. The plant mitoribosome thus also significantly diverged from plastid ribosomes that are comparatively much more like bacterial ribosomes[71].

Phylogeny studies, i.e., of the distribution of PPR proteins[29], suggest that the plant-specific features observed here in cauliflower mitoribosomes are conserved in the entire group of angiosperms. Likewise, the plant-specific process, involving RsgA, to prevent mRNA association before the completion of the ribosome small subunit maturation, is anticipated to be conserved at least in dicotyledon plants.

Altogether, this study advances our understanding of the plant mitochondrial translation apparatus structure and function, revealing specific features that distinguish plant mitoribosomes from their counterparts in other eukaryotes as well as in prokaryotes. These results show that while plant mitoribosomes have followed a distinct evolutionary path, they have preserved bacterial-like rRNA modifications in their comparatively well-conserved core domains while also incorporating unique plant-specific proteins and evolving specific assembly mechanisms. These structural features underscore the functional complexity and evolutionary adaptations of plant mitoribosomes, possibly required to fulfill plant-specific functions, such as decoding a set of mRNAs that are more diverse and more complex than in other eukaryotes.

## Methods

### Mitochondria and ribosome purification

Mitochondria from cauliflower were purified as previously described [13,20,29]. For the chloramphenicol-stalled mitoribosomes, mitochondria purification was carried out in the presence of 200 μg/mL chloramphenicol. Mitoribosome purification was conducted as previously[72,73]. In brief, purified mitochondria were re-suspended in Lysis buffer (20 mM HEPES-KOH, pH 7.6, 70 mM KCl, 20 mM MgCl₂, 1 mM DTT, 1.6% Triton X-100, supplemented with proteases inhibitors (Complete EDTA-free)) to a 1.3 mg/ml concentration and incubated for 15 min in 4 °C. Lysate was clarified by centrifugation at 25,000 g, 20 min at 4 °C. The supernatant was loaded on a 40% sucrose cushion in Monosome buffer (Lysis buffer with 0.1% Triton X-100) and centrifuged at 235,000 g, 3 h, 4 °C. The crude ribosome pellet was re-suspended in Monosome buffer and loaded on a 10-30 % sucrose gradient in the same buffer and run for 16 h at 65,000 g. Fractions corresponding to mitoribosomes were collected, pelleted and re-suspended in Monosome buffer and directly plunge frozen for cryo-EM analysis. For the samples treated with chloramphenicol or GTP analog, the same procedure was used, but with 200 μg/mL chloramphenicol or 200 μM Guanosine-5'-(γ-thio)-triphosphate, Tetralithium salt, (GTPγS).

### Preparation of total RNA and DNA from purified mitochondria

Total mitochondrial RNA was extracted from purified cauliflower mitochondria pellet using TRI reagent (Sigma-Aldrich). After isopropanol precipitation, the RNA pellet was washed with 70% ethanol, air-dried, and resuspended in nuclease-free water. Total RNA was then treated with Turbo DNase (Thermo Fisher Scientific), purified with phenol/chloroform, and further subjected to RNA Clean and Concentrator kit (Zymo Research). The total mitochondrial RNA was used for direct RNA sequencing (DRS) library preparation. Total mitochondrial DNA was extracted from purified cauliflower mitochondria pellet using TRI reagent (Sigma-Aldrich). The sequences of 26S and 18S rRNA genes were introduced in a plasmid vector, that was used as a template for their subsequent amplification. The rRNA genes were amplified with primers 26S_F/26S_R and 18S_F/18S_R for 26S and 18S, respectively. The sequence of the T7 polymerase was introduced to the 5' of the amplified products via their forward primer. In vitro transcription was performed using the T7 RiboMAX Large Scale System (Promega), according to the manufacturer's instructions. The reactions were subsequently treated with DNase (Promega), purified with phenol/chloroform, and further subjected to MicroSpin G-50 columns (Cytiva). The in vitro transcribed 26S and 18S rRNAs were used for DRS library preparation.

### Direct RNA sequencing library preparation

For DRS library preparation, custom reverse transcription adapters (RTAs) containing Deeplexicon multiplexing barcodes (BC1, BC2 or BC3)[74] were designed for sequence-specific ligation to the 3'-ends of B. oleracea mitochondrial rRNAs (Supplementary Table 5). For native rRNAs, BC1 was included in the RTA oligos (Oligo A and Oligo B), whereas for each IVT rRNA, a different barcode sequence was assigned: BC3 for IVT 26S and BC2 for IVT 18S. Oligo A and Oligo B for each rRNA target were annealed (1:1) at 2 mM in annealing buffer (10 mM Tris-HCl, pH 7.5, 50 mM NaCl) by heating to 95 °C for 2 min and letting them cool down gradually (0.1 °C/sec). The libraries were prepared following the Oxford Nanopore Technologies (ONT) DRS protocol for PromethION FLO-PRO002 R9.4.1 flow cells using reagents from the DRS kit (SQK-RNA002, ONT) and the custom RTAs. Briefly, 1 μg of total mitochondrial RNA or 2 pmol of each IVT rRNA were ligated with their respective custom RTA using T4 DNA ligase (Thermo Fisher Scientific) for 15 min at room temperature, before they were reverse transcribed with SuperScript III Reverse Transcriptase (Thermo Fisher Scientific). The RNA:DNA hybrid products were purified with 1.8× Agencourt RNAClean XP beads (Thermo Fisher Scientific), washed with 70% ethanol, and ligated with the RNA Adapter (RMX). The products were purified with 0.4× Agencourt RNAClean XP beads, washed twice with wash buffer (WSB), and eluted in elution buffer. Finally, the RNA library was mixed with RNA Running Buffer (RRB) and loaded onto a primed FLO-PRO002 flowcell (ONT).

### Basecalling and demultiplexing of ONT direct RNA sequencing data

The ONT output from three DRS runs performed on two distinct FLO-PRO002 flowcells (Supplementary Table 6) were processed via *deeplexicon_sub.py* dmux (Deeplexicon v1.2.0, options: -f multi -s 0.95 -m models/resnet20-final.h50)[74] on an Nvidia V100 GPU accelerator to obtain demultiplexing calls for all the fast5 reads. These reads were basecalled with Guppy (v6.5.7, config: "rna_r9.4.1_70bps_hac_prom.cfg") and the fastq output split into separate files (BC1, BC2, BC3, BC4). The demultiplexed and basecalled ONT reads were filtered by length (2000-3300 nt for 26S and 1700-2000 nt for 18S reads) and then aligned to the B. oleracea rRNA reference sequences in Geneious (v2024.0.7) using "Map to Reference" with the built-in mapper. Near full-length reads were selected for current intensity analysis to determine whether nucleotide modifications observed by cryo-EM could be validated by ONT DRS. Base-calling error was calculated for each nucleotide as a percentage of the highest non-reference base (observed in native rRNA data compared to the rRNA reference sequence) separately for in vitro *transcribed* (IVT) RNA (Error

IVT, %) and native RNA (Error Native, %). Then, the per-base difference between these native and IVT error values (ΔCallError, %) was computed to identify positions with putative modifications. Positions with a ΔCallError > 10% were considered indicative of RNA modifications. This 10% threshold corresponds to approximately three times the average background miscalling level (2.9%).

## Analysis and visualization of current intensities

Raw fast5 files were basecalled again using Guppy (v6.3.8, options: "--kit SQK-RNA002 --flowcell FLO-PRO002 –fast5_out") to include FastQ entries, Move and Trace tables, which are necessary for the current intensity analysis. Full-length reads were selected from the fastq file and mapped with minimap2 (v2.26, options: "-ax splice -uf -k14") to their corresponding reference sequences. To extract event-level information from these basecalled fast5 files, we used f5c (v1.5, https://github.com/hasindu2008/f5c)[75], a GPU-accelerated implementation of Nanopolish (https://github.com/jts/nanopolish) that aligns ONT signals called "events" to reference k-mers. First, we indexed the FastQ file using the command *f5c index* to link each read to its basecalled fast5 file. Next, we extracted per-read current intensities using *f5c eventalign* with the options "--rna --print-read-names --collapse-events -b $bam -g $ref_sequence -r $fastq > f5c_eventalign.tsv". The resulting alignment and eventalign output files were then analyzed in R. Namely, we used the *pileup* function from the Rsamtools package (v2.20.0) to obtain nucleotide frequencies for each position in the alignments. We then incorporated eventalign information to display the nanopore-derived current intensities at specific nucleotide positions. All visualizations were performed using ggplot2 (v3.5.1).

## Mass spectrometry analyses of RNA modifications

Chemicals used were of analytical grade or high purity unless indicated otherwise. Water used to prepare sample solutions or buffers was obtained from Merck Millipore. Ammonium acetate ($NH_4OAc$), triethylamine (TEA), ZipTip C18 and acrylonitrile were purchased from Sigma-Aldrich, and 1,1,1,3,3,3-hexafluoropropan-2-ol was purchased from Honeywell. RNase T1, RNase A and RNase H were purchased from Thermo Fisher Scientific. RNase 4 were purchased from New England Biolabs. RNase MC1 and cusativin were provided by Waters. Oligonucleotides used for RNA–DNA hybridization were synthesized by IDT.

For rRNA isolation, the 26S and the 18S rRNA were isolated using a gel filtration column (Superose 6 Increase 10/300 GL column) by isocratic elution with 300 mM $NH_4OAc$ at 0.5 ml min$^{-1}$ on an ÄKTA purifier system (General Electric Healthcare). The specific cleavage of rRNA was done as described[76,77]. Prior to RNase H cleavage, 10 µg rRNA and 10 µg oligonucleotides were mixed and heated at 80 °C for 2 min followed by a cooling down to 50 °C for DNA–RNA hybridization (list of oligonucleotides in Supplementary Table 5). DNA–RNA duplexes were digested at 50 °C for 30 min by adding 1 µl of 10× RNase H buffer, then 2.5 µl of 0.5 U µl$^{-1}$ RNase H. After RNase H cleavage, the fragments of interest were isolated by 10% denaturing polyacrylamide gel. Gels were run at 16 W for about 4 h. After migration, gels were stained with ethidium bromide, and the bands containing the fragments of interest were visualized and excised under ultraviolet light. Bands were dried under vacuum and stored at −20 °C.

To detect pseudo-uridines, RNase H fragments were subjected to in-gel cyanoethylation following the procedure described by[78]. Briefly, gel pieces were treated by 30 µl of 41% EtOH/1.1 M triethyl-ammonium acetate (TEAA) (pH 8.5) and 3 µl of acrylonitrile. Specific labeling of pseudo-uridines was performed for 2 h at 70 °C. After derivatization, the supernatant was removed, and the gel pieces were washed three times with 200 mM $NH_4OAc$. Pieces of gel were then dried under vacuum and stored at −20 °C.

Dried pieces of gel containing RNA were digested with the appropriate enzyme. For RNase T1 or RNase A, 20 µl of enzyme (1/1,000 in 100 mM $NH_4OAc$) were added. Samples were then incubated at 55 °C for 2 h. In-gel digestions by RNase 4 were carried out using 3 units/piece of gel in 100 mM $NH_4OAc$ at 37 °C for 60 min. The MC1 digestions were carried out using 7.5 units/piece of gel in 200 mM $NH_4OAc$ (pH 8.5) at 37 °C for 60 min. The cusativin digestions were carried out using 7.5 units/piece of gel in 200 mM $NH_4OAc$ (pH 9.0) at 37 °C for 60 min. After digestion, the samples were desalted with ZipTip C18 using 200 mM $NH_4OAc$. Eluates were dried under vacuum, and pellets containing RNase digestion products were resuspended in 3 µl of milliQ water before injection. Separation of products were achieved on an Acquity peptide BEH C18 column (130 Å, 1.7 µm and 75 µm × 200 mm) using a nanoAcquity system (Waters). The analysis was performed with an injection volume of 3 µl. To avoid RNase contamination, the chromatographic system was thoroughly washed when analyzing samples digested with another RNase. The column was equilibrated with eluant A containing 7.5 mM TEAA, 7.0 mM TEA and 200 mM 1,1,1,3,3,3-hexafluoropropan-2-ol at a flow rate of 300 nl min$^{-1}$. Oligonucleotides were separated using a gradient from 15% to 35% of eluant B (100% methanol) for 2 min, followed by an increase of eluant B to 50% in 20 min. The column oven temperature was set at 65 °C. Detection by MS and tandem MS was achieved using SYNAPT G2-S from Waters Corporation. The source settings used included polarity mode: negative ion; capillary voltage: 2.6 kV; sample cone voltage: 30 V; and source temperature: 130 °C. The samples were analyzed over an m/z range from 550 to 1600 for the full scan, followed by a Fast data direct acquisition scan with an m/z range from 200 to 2000. Collision-induced dissociation experiments were achieved using Argon gas.

## Cryo-EM grid preparation

4 µL of purified ribosome at a protein concentration (OD280) of 3 µg/µL was applied onto Quantifoil R2/1 200-mesh holey carbon grids, pre-coated with a 2 nm continuous carbon film and glow-discharged (2.5 mA for 20 s). Samples were incubated on the grid for 25 s and then blotted with filter paper for 2 s in a temperature and humidity controlled Vitrobot Mark IV (T = 4 °C, humidity 100%, blot force 5) followed by vitrification in liquid ethane.

## Single particle cryo-EM Data collection

Data collection was performed using a 300 kV G4 Titan Krios electron microscope (ThermoScientific) equipped with Falcon4i camera and a Selectris X energy filter using EPU for automated data acquisition at the IGBMC's Integrated Structural Biology platform. Data were collected at a nominal underfocus of −0.5 to −2.5 µm at a magnification of 165,000x, yielding a pixel size of 0.729 Å.

For the monosomes, two datasets (1 and 2) were recorded as movie stacks, from 2 different grids of 9091 and 7825 micrographs, respectively and exported as EER files. Each movie stacks were fractionated into 40 frames for a total electron dose of 41.66 e$^-$/Å$^2$. For the small subunit dataset, 20,145 micrographs were recorded as movie stacks and exported as EER files. Each movie stacks were fractionated into 40 frames for a total electron dose of 50.96 e$^-$/Å$^2$.

For the stalled monosomes with P-site tRNA, data collection was performed using a 300 kV Titan Krios electron microscope (ThermoScientific) equipped with a K2 camera and a GIF energy filter using SerialEM for automated data acquisition at the Biozentrum's BioEM facility. Data were collected at a nominal underfocus of −0.5 to −2.5 µm at a magnification of 130,000×, yielding a pixel size of 1.058 Å. 5935 micrographs were recorded as movie stacks, exported as TIF files. Each movie stacks were fractionated into 50 frames for a total electron dose of 46 e$^-$/Å$^2$.

## Single particle cryo-EM Data processing

For all datasets, the entire processing pipeline was carried out in cryoSPARC[79], and the processing workflows are shown Supplementary Figs. 1–4. Cryo-EM validation parameters are indicated in Supplementary Table 1.

For the monosome datasets, pre-processing and particle picking were performed independently, and the resulting good particles were then merged. For dataset 1, after motion correction and CTF estimation, 8829 micrographs were kept, and 7532 micrographs were kept for dataset 2. For particle picking and 2D classification, the same approach used for the small subunit dataset was used, but with particles extracted with a box size of 756 pixels, down-sampled to 256 pixels (pixel size of 2.1528). For dataset 1, 137,759 particles remained after 2D classification and then 111,531 after initial 3D classification. For dataset 2, 205,994 particles remained after 2D classification and then 174,386 after initial 3D classification. At this point, particles were merged, resulting in a total number of 258,917 particles that were further cleaned down to 219,550 particles. They were re-extracted for high-resolution refinement, using a box size of 756 pixels, down-sampled to 560 pixels (pixel size of 0.9841). A global refinement reached 2.77 Å resolution. After Global and Local CTF refinement, the resolution improved to 2.34 Å. Reference Based Motion Correction was then performed, and further improved the resolution to 2.18 Å. Another round of Global and Local CTF refinement at full resolution (bin1) improved the resolution to 2.11 Å. Particles were down-sampled to 588 pixels (pixel size of 0.9373), and focused refinement were performed using a mask for the LSU (1.99 Å), and then further local refined for the LSU L7/12 Stalk (2.85 Å), the Central Protuberance (2.09 Å) and the LSU's back (2.53 Å). The same was done for the SSU (2.35 Å), and then SSU head (2.24 Å), head protuberance (2.41 Å), SSU body (2.25 Å), foot (2.47 Å) and body protuberance (2.58 Å). To improve the SSU head extension, signal subtraction (with a mask to remove the signal of most of the ribosome) and particle recentering was performed, allowing to resolve the head extension to 3.87 Å and then further to 3.43 Å for the base, 4.04 Å for the core and 4.32 Å for the tip.

For the small subunit dataset, movies were motion corrected, and CTF was estimated using Patch Motion Correction and Patch CTF. Bad micrographs were discarded, resulting in 19,932 out of 20,145 micrographs remaining. Particles were then picked using a template picker approach, yielding 1,896,051 initial positions that were extracted with a box size of 720 pixels, down sampled to 280 pixels (pixel size of 1.8746) for faster processing. Particles were cleaned iteratively through 2 rounds of 2D classification, from which 931,475 particles were selected. Bad particles were then further discarded in 3D through a combined approach of Ab-initio, Heterogeneous Refinement and 3D Variability analysis, from which 602,798 particles remained. Good particles were then further sorted using a mask covering the body of the SSU, resulting in 2 main classes: an intact empty small subunit class (388,094 particles), and the RsgA class (208,855 particles). At this point, particles were re-extracted for high-resolution refinement, using a box size of 720 pixels, down-sampled to 560 pixels (pixel size of 0.9373). A global refinement of the empty small subunit class reached 2.3 Å resolution, while the RsgA class reached 2.44 Å resolution. After Global and Local CTF refinement, resolution improved to 2.05 Å and 2.25 Å, respectively. Reference Based Motion Correction was then performed, and further improved the resolution to 2.04 Å and 2.27 Å, respectively. Another round of Global and Local CTF refinement was performed at bin1. Particles were down sampled to 588 pixels (pixel size of 0.8927), and then focused refinements were performed using masks for the SSU head and body and for RsgA, resulting in, for the RsgA class, 2.07 Å for the RsgA focused map, 2.17 Å for the body and 2.07 Å head.

For the stalled monosomes, 5621 micrographs were kept after motion correction and CTF estimation. Particles were picked and extracted with a 564 pixels' box size, down sampled to 282 pixels (pixel size of 2.1160) and cleaned through 2 rounds of 2D classification, resulting in 295,205 particles. After cleaning the particles in 3D, 219,458 particles were selected and submitted to focused 3D classification with a mask covering the decoding center, made from A, P and E site tRNA models. This yielded a clear class of P-site tRNA stalled monosomes with 34,964 particles. For high-resolution refinement, particles were re-extracted using a box size of 500 pixels (pixel size of 1.1934). After Global and Local CTF refinement, the resolution reached 3.17 Å. Reference Based Motion Correction was then performed, and further improved the resolution to 3.05 Å. Another round of Global and Local CTF refinement at full resolution (bin1), and then focused refinements were performed using masks for the LSU and SSU that respectively reached 2.95 Å and 3.08 Å.

## Model building and refinement

Even though a model for the plant mitoribosome was already available (PDB: 6XYW), we decided to build all the proteins de novo, taking a completely unbiased approach and making the most of these high-resolution maps. For that, focused, refined cryo-em maps (all at resolution better than 3 Å) for the different main parts of the mitoribosome (large subunit, body and head of the small subunit) were used as inputs for ModelAngelo[32] without input sequence. Peptide chains obtained were then blasted against the Brassicaceae taxon (3700) of the UNIPROT database, resulting mainly in hits from *Arabidopsis thaliana*, and *Brassica oleracea var. oleracea* and the corresponding AlphaFold2 models were retrieved for all the identified proteins and then matched to the ModelAngelo models using the Matchmaker tool in ChimeraX[80], which were further rigid-body fitted in their respective cryo-EM maps. The protein models were then manually inspected and refined in COOT[81], where the best fitting protein, sequence-wise, were chosen. For the rRNAs, the models from 6XYW were rigid-body fitted in the density and then refined with restraints in COOT. rRNA extensions in lower resolutions areas (e.g., the head extension of the SSU or back extension of the LSU) were modeled just as stretches of A and U. The sequences were then mutated accordingly and refined again. Ions, ligands and rRNA modifications were first placed by homology with *E. coli* 8B0X and then manually curated in COOT. Finally, water molecules were placed using the PHENIX phenix.douse tool and curated using the "Check Waters" tool in COOT. The different parts of the model were then automatically refined in PHENIX[82] against the best resolved focus-refined maps using the phenix.real_space_refine tool and then again manually refined in COOT, this repeating through several cycles. Chimeric maps were generated by using the Map Box option in PHENIX to cut out densities around the models (3.5 Å around the atoms). Maps were then fitted into the consensus maps in ChimeraX and combined using the 'vop maximum' command. The model geometry was validated using MolProbity[83].

## Figure preparation

All figures were prepared using UCSF ChimeraX, developed by the Resource for Biocomputing, Visualization, and Informatics at the University of California, San Francisco, with support from National Institutes of Health R01-GM129325 and the Office of Cyber Infrastructure and Computational Biology, National Institute of Allergy and Infectious Diseases[84,85].

## Reporting summary

Further information on research design is available in the Nature Portfolio Reporting Summary linked to this article.

# Data availability

The single particle cryo-EM maps of *B. oleracea* mitoribosome have been deposited at the Electron Microscopy Data Bank (EMDB) and models on the protein data bank (PDB). Full high-resolution mitoribosome EMD-151718 (PDB 9GYT), unfocused EMD-51703, focused LSU EMD-51710, focused LSU CP EMD-51711, focused LSU L7/12 stalk EMD-51712, focused LSU back extension EMD-51713, focused LSU rPPR5 EMD-51714, focused SSU body EMD-51704, focused SSU body protuberance EMD-51709, focused SSU foot rPPR1 EMD-51708, focused SSU foot rPPR10 EMD-51707, focused SSU head EMD-51705, focused SSU head S3 area EMD-51706, focused SSU head extension base EMD-51715,

focused SSU head extension core EMD-51716, focused SSU head extension tip EMD-51717. Small subunit in presence of RsgA EMD-50014 (PDB 9EVT), focused SSU head EMD-50015, focused SSU body EMD-50017, focused RsgA EMD-50016. P-site tRNA mitoribosome stalled with chloramphenicol EMD-50011 (PDB 9EVS), focused LSU EMD-50012, focused SSU EMD-50013. The Oxford Nanopore DRS datasets are available at the Zenodo database https://doi.org/10.5281/zenodo.14196969.

## Code availability

The Oxford Nanopore DRS bioinformatics code for the detection of rRNA modifications is available at the Zenodo database https://doi.org/10.5281/zenodo.14196969.

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

## Acknowledgments

We thank M. Chami and D. Kalbermatter from the University of Basel - Biozentrum BioEM facility for their help in operating the electron microscopes. We thank Prof. Dr. Benjamin D. Engel for hosting Dr. Florent Waltz at the Biozentrum. We thank Dr. Asier González Seviné and

Prof. Dr. Mihaela Zavolan for their access to the BioComp gradient fractionator and Dr. Irene Vercellino for advice on generating chimeric cryo-EM maps. This work used the Integrated Structural Biology platform of the Strasbourg Instruct-ERIC center IGBMC-CBI, supported by FRISBI (ANR-10-INBS-0005). We thank Dr. Alexandre Durand for his help in operating the electron microscopes. We thank the Strasbourg Esplanade proteomic platform for the proteomic analysis. This work was supported by the "Center National de la Recherche Scientifique", the University of Strasbourg, by Agence Nationale de la Recherche (ANR) grants 'DAMIA, ANR-20-CE11-0021' and 'PROPHAN, ANR-22-CE12-0008-01' to PG and by the LabEx consortium "MitoCross" in the frame of the French National Program "Investissement d'Avenir" ANR-11-LABX-0057_MITOCROSS. This work was supported by the Swiss National Science Foundation (SNSF) with a Swiss Postdoctoral Fellowship (number 210561), an Ambizione Grant (number 216094) and by the Alexander von Humboldt Foundation through the Humboldt Research Fellowship Program for Postdocs to FW. P.W. was supported by the French National Program Investissement d'Avenir (Labex NetRNA) administered by the Agence Nationale de la Recherche (ANR-10-LABX-0036_NETRNA). This work of the Interdisciplinary Thematic Institute IMCBio, as part of the ITI 2021-2028 program of the University of Strasbourg, CNRS and Inserm, was supported by IdEx Unistra (ANR-10-IDEX-0002), and by SFRI-STRAT'US project (ANR 20-SFRI-0012) and EUR IMCBio (ANR-17-EURE-75 0023) under the framework of the French Investments for the Future Program.

## Author contributions

V.S., T.T.N., and N.C. performed biochemical purifications. F.W. and V.S. performed cryo-EM data processing, model building and interpretation, and initial writing of the manuscript. V.S., D.P., and T.B. performed nanopore analyses. P.W. performed rRNA MS analyses. Y.H. contributed to the writing of the manuscript. F.W. and P.G. conceived the project, interpreted results, and finalized the manuscript.

## Competing interests

The authors declare no competing interests.
