## [Transparent Peer Review file · Nature Communications]

Structural insights into cauliflower mitoribosome in translation state and in association with a late assembly factor

Corresponding Author: Dr Florent Waltz

Version 0:

Reviewer comments:

Reviewer #1

(Remarks to the Author)

Compared to the structure they published four years ago (Waltz et al., 2020), the authors have now improved the resolution to ~2.2 Å. These improved maps allowed the identification of additional ribosomal proteins that were previously unassigned, as well as universal and plant-specific modifications. Furthermore, the authors resolved a chloramphenicol-stalled ribosome with a P-site tRNA and an immature SSU with the assembly factor RsgA.

While the high-resolution model of the cauliflower mitoribosome could serve as a valuable reference structure for future studies, the manuscript in its current form is not ready for publication. I have several major concerns that need to be addressed.

1. The assignment of new ribosomal proteins is not particularly interesting, especially since these proteins were already present in the previous low-resolution maps. Are these newly assigned proteins specific to plants? If so, the focus should be placed on those proteins. Otherwise, this section should be significantly reduced. A single figure comparing all the newly assigned proteins with the previous structure would be sufficient.
2. The same applies to the chloramphenicol-stalled ribosome. This structure does not provide novel insights into how plant mitoribosome translation occurs. Are there any distinct structural features compared to known mitoribosome structures? If not, this section should also be shortened. If the authors aim to discuss translation states, the research should be focusing on untreated polysomes instead.

Overall, I recommend reorganize these two parts and shifting the focus toward the plant-specific modifications and the RsgA structure. While this would require substantial rewriting, it would greatly enhance the impact and value of the manuscript.

3. Although cauliflower and Arabidopsis belong to same group, it is very important to build your model basing on the cauliflower genome being used. If the genome is not well annotated, it is necessary to annotate it yourself (collaboration or Biotech company?). Since the authors aim to provide improved models, the precise sequences (proteins and rRNAs) are obligatory. This must be corrected.
4. It is surprising that the structural and nanopore-identified modification sites are inconsistent. Modifications observed in the structure, especially methylation, should be reliable given the map resolution they achieved. This inconsistency suggests that nanopore sequencing may not be an optimal method for detecting modification sites in this case. The authors should verify these modification sites, particularly the interesting cauliflower-specific ones, using additional classical experimental methods.
5. The title needs to be revised. The current title is vague and uninformative. It should be both specific and informative, explicitly mentioning the cauliflower mitoribosome. Meanwhile in the main text, using "flowering plants mitoribosome" is misleading and should be specific.
6. Does RsgA exist in fungi or animal mitochondria, or is it specific to plants? Clarifying this point would add value to the manuscript. Additionally, comparing the RsgA structure with the known bacterial structure would provide insight into what is new and important in this study. In Arabidopsis thaliana, there are two isoforms of RsgA: RsgA1 and RsgA2. Notably, only RsgA1, which is used in the model, contains an extra C-terminal extension. What is the situation in cauliflower? Once again, please build the cauliflower model.

Here are some suggestions to improve the manuscript:

1. The abstract needs significant revision. For instance, the sentence “However, the molecular mechanisms of translation by plant mitoribosomes remain largely unknown” does not align with the content of this manuscript, as this problem is not addressed here. Additionally, the use of “extensive” to describe the modifications is misleading, as only dozens of modified sites were identified. Please remove or revise this phrasing in the abstract and throughout the manuscript.
2. In line 82, specify which sample and dataset the described structures are derived from.
3. In Figure 2a, please show all (also could be in supplementary figures) the identified modifications supported by density maps to provide comprehensive visualization.
4. In Figure 3b, the nascent chain appears suspicious as it seems disconnected from the CCA end. In the human mitoribosome, a mitoribosome protein binds in the peptide exit tunnel and reaches the PTC. Please check if such a protein is present in cauliflower mitoribosome.
5. Figure 3e, the inset is not prominent and difficult to catch. Consider adding a frame around the inset to improve clarity. In general, please avoid using white shading to separate panels.
6. Figures S5 and S6, the overall views do not help. Use consistent colors for all proteins and label them properly for better comprehension.
7. For the data processing section, provide the model-to-map FSC and include model resolution statistics in Table 1. Additionally, for Table 1, remove statistics for focus-refined maps as they do not add value and make the table harder to read.
8. The interaction between m2A2808 and A2377 mentioned in line 206, as well as the two PPR modules described in line 132, should be illustrated with additional figures.
9. The methodology for GTPγS treatment is unclear, especially since it is hard to believe it was done in the same way as chloramphenicol treatment. Furthermore, the chloramphenicol concentration is inconsistent—500 μg/mL or 200 μg/mL?

(Remarks on code availability)

Reviewer #2

(Remarks to the Author)

The authors of the manuscript entitled „Structural insights into maturation and translation of a plant mitoribosome” seek to structurally characterize the mitoribosome of flowering plants – they purify their samples from cauliflower. They use cryo-electron microscopy (cryo-EM) to determine the three-dimensional structures of resting and translationally stalled mitoribosomes at very high overall resolutions (up to 2.1 Å). In addition, they imaged a late assembly stage of the small subunit in complex with an assembly factor (i.e. RsgA). This manuscript mainly describes structural observations, but the presence and identity of RNA modifications was additionally verified using nanopore sequencing. The presented structures were determined at much higher resolution than the previously available structures for plant mitoribosomes – either from *A. thaliana* or cauliflower. Thanks to the significantly improved resolution, the authors were not only able to identify new core ribosomal proteins and RNA features, but also to see several RNA modifications. The authors thoroughly list and describe a long list of novel features of the plant mitoribosome, which differ them from other eukaryotic (e.g. yeast and mammalian) mitoribosomes and bacterial ribosomes.

I think that the work significantly expands our knowledge of plant mitoribosomes, and how they compare to other (mito)ribosomes. Therefore, I find reviewed manuscript suitable for the publication in Nature Communications, provided that the authors resolve the two main issue and consider addressing the listed minor issues. Of note, I do not question the scientific validity of the presented results but would like the authors to improve the clarity of the manuscript for a broad audience.

Major issues

1. To improve the map quality in regions of particular interest and of newly identified components, the authors performed several local refinement steps. Subsequently, they used these locally refined maps to generate composite maps – such types of maps are sometimes called “Frankenstein maps”. The usefulness of such local refinement steps is undisputed, but the generation of composite maps is heavily debated in the field. I don’t see any advantage of using such artificially stitched maps over globally refined maps. Showing locally refined maps can of course be used to zoom into the locally refined area. I understand the temptation to generate prettier maps for prettier figures but combining several locally refined maps into a composite map (e.g. Fig. 1a/b, Fig3a and Fig.4a) does not relate directly to the obtained data. I would encourage the authors to replace those maps with the obtained global maps. At the obtained resolution, the maps will still look nice, but it will be become more obvious for the non-experts, where the author identified dynamic movements.
2. I don’t fully understand why the authors would not show the densities for all modified rRNA nucleotides. They should also explain to the reader, how they are able to discriminate between U and Pseudo-U. I would suggest expanding Figure 2a by showing the densities for all modified nucleotides in the LSU. In addition, I would encourage the authors to present the densities for all positions that were detected by nanopore sequencing, which could not be confirmed by cryo-EM – I guess, this could be shown a Suppl. Figure. I am particularly curious about the density around SSU position 959. This would allow the reader to understand the rational behind the decision of the authors to call or not to call a modification at a certain position. I would also like to hear the opinion of the authors for those modifications that can be detected by cryo-EM, but not by Nanopore sequencing. At the moment, the whole RNA-modification field seems to be excited about Nanopore sequencing, but apparently some highly modified sites are not detected – this kind of information about the weakness of nanopore-based detection should reach the interested expert reader.

Minor issues

- Typically, ribosomes are purified either from the monosome or polysome fractions. For people from outside the ribosome

field an additional supplementary panel or figure explaining the sample preparation process would help.

- In Supplementary Table 1 there is no magnification value specified for the 9EVS structure – please complete. Actually, it states “???” which does not look very scientific.
- Fig. 1a and 1b – both panels are not referenced in the main text of the manuscript.
- Fig. 1c - please provide labels for each of the visible ions.
- Fig. 1e panel is introduced in the text before Fig. 1d.
- Fig. 2c - please consider changing the grey color of labels above grey background.
- Fig. 2c legend – missing space “.. grey.”?” ..”.
- Fig. 3a and 3c panels are not referenced in the main text of the manuscript.
- Fig. 3d panel is referenced in the main text before Fig. 3b.
- Labels SSU/LSU in Fig. 3a have different colors than the same labels in Fig. 2. Please consider improving the consistency between main and supplementary figures i.e. same colors for mitoribosome components in all figures.
- Fig. 3b - Potassium ions should be labeled.
- Page 6 - Consider rephrasing “peptide exit via the peptide tunnel” to “...the end of exit tunnel”
- The label “d” in Fig. 4a should be highlighted or removed.
- Fig. 4b panel is not referenced in the main text of the manuscript.
- The panel referencing order of Fig. 4 in the main text is a, g, d, e and c.
- For Supplementary Fig. 1-4 please consider pipelines simplification, specify types of particle picking, remove redundant 2D averages, show scales of 2D averages, for each density include orientational bias, show all used classification masks, label each local refinement mask
- Use 2 colors for the gradient in the local resolution estimations – here white is higher res than red, which might be misleading to the non-experts. Local resolutions should be calculated for the same FSC cutoff as reported GSFSCs (0.143).
- Some weird box-artefact in Suppl. Fig. 3b can be observed
- Suppl. Fig. 5a - bL25-2m has different color than the same protein in Fig. 1e.
- Suppl. Fig. 5c – maybe H4 should also be in bold.
- Labels for p56 and p58 are missing in Suppl Fig. 5d.
- In Supplementary Fig. 6a legend “.. of domain I, forming p17, p17 and p21-22, specific ..” should rather be “.. of domain I, forming p15, p17 and p21-22, specific ..”...but maybe I misunderstood.
- Suppl. Fig. 7c - no label on the panel for uS2m.
- Suppl. Fig. 7c – label for mRNA exit channel can hardly be read – consider reformatting.
- Did the authors analyzed also the large subunit fraction isolated from the mitoribosomes purified in the presence of non-hydrolysable GTP analogue to see whether any additional factors are binding?
- Supplementary Table 1 is not referenced in the main text of the manuscript.
- In Supplementary Table 1 local resolutions should be specified for FSC=0.143, as indicated for global resolution.
- Supplementary Table 4 should also be referenced in the main text of the manuscript, not only in the legend of Fig. 2.
- Supplementary Fig. 6c panel is referenced in the main text before Supplementary Fig. 6a and 6b.
- Supplementary Fig. 8b panel is referenced in the main text before Supplementary Fig. 8a.
- Supplementary Fig. 9 and 10 should also be referenced in the main text of the manuscript, not only in the Fig. 2 legend.

Of note – during my search, I got confused whether the previous PDB entry 6XYW shows the cauliflower or *A. thaliana* mitoribosome. The paper clearly relates to cauliflower, but the PDB entry list *A. thaliana* as the organism...could the authors resolve this confusing issue with the PDB and EMDB?

(Remarks on code availability)

Reviewer #3

(Remarks to the Author)

(Remarks on code availability)

Reviewer #4

(Remarks to the Author)

In the study, high resolution Cryo-EM structural data were collected for the mitochondrial ribosome of cauliflower. The data were collected from ribosomal particles in three different states, including ribosomes with bound tRNA at the P-site and intermediate assembly states. Due to the high quality of the data, the authors were able to identify new sections, which were previously not resolved, e.g. densities of additional ribosomal proteins. Interestingly, rRNA modifications could be identified, which were previously not described for mitochondrial ribosomes.

The dataset seems mature and of high quality and the figures are very well designed. I cannot evaluate the technical details of the structural data, but I did not find major problems within the text, which require major revision.

However, by reading the manuscript it is not fully clear to me how these structural data provide new insights that would really

justify publication in the submitted high rank journal. Previously, several structural data of mitochondrial ribosomes from algae, Arabidopsis and cauliflower contributed to a good and high-resolution understanding about mitochondrial ribosomes. This also includes work from the same group. Certainly, the findings about the rRNA modifications and the assembly intermediate states are interesting. However, it needs to be better elaborated how the discovered features really contribute to a deeper and broader understanding about ribosome biology in plant mitochondria. The text reads very technical and needs a broader perspective. By reading the current version, several questions arise: Which of the unknown aspects are now solved? The way it is written, it appears that the findings are just additions to the known features (although the authors tried to mention this but this reads very vague). Are these new findings specific to cauliflower mito ribosomes? Which features are common for plant cells and how do the data help to generally understand specific features of (plant) mito ribosomes? How do these ribosomes functionally differ from 80S ribosomes, bacterial or plastidic 70S ribosomes. In the results and discussion chapter, the authors only briefly tried to compare their findings to other ribosomal structures. I would support publication in the journal if these points can be clarified.

(Remarks on code availability)

Version 1:

Reviewer comments:

Reviewer #1

(Remarks to the Author)

We highly appreciate that the authors have rebuilt their structural models based on the annotated genome of *Brassica oleracea* var. *botrytis* and complemented their findings with mass spectrometry to confirm rRNA modification sites. The manuscript has also been substantially rewritten to highlight plant-specific features. Nevertheless, several issues require additional clarification.

1. In the section "Cauliflower mitoribosome contains a set of at least 20 rRNA modifications", the phrase "at least" remains ambiguous. We expect the authors to clearly define which modifications are robustly supported, so that readers can confidently interpret the data. It is not sufficient to present all detected sites and leave it to the reader to determine which are reliable. Examples are 891, 1105, 2762, 2803, they are shown in the main figure, should we trust them? As a suggestion, in Supplementary Figure 11, please label which peaks correspond to modified signals in the MS data.
2. In the section "Identification of plant-specific features of the translating mitoribosome", although the authors have addressed the reviewer's points in their rebuttal, the plant-specific features are still not easily to get for readers who are not experts in the mitoribosome field. To improve clarity, we suggest adding structural comparisons or schematic overlays with other mitoribosomes to clearly highlight the unique plant-specific features.
3. The UniProt ID F4HTL8 corresponds to RsgA2 in *Arabidopsis thaliana*, which lacks the C-terminal extension. It remains unclear whether homologs in *Brassica oleracea* all possess such an extension.
4. There is inconsistency in the naming of the bL25m isoform, referred to as "bL25m-2" in Line 115 but as "bL25-2m" in Figure 1 and Supplementary Figure 5b.

(Remarks on code availability)

Reviewer #2

(Remarks to the Author)

I have reviewed the strongly improved version of the revised manuscript by Skaltsogiannis and colleagues. I still disagree with the display of the composite maps, but as the authors clearly indicate the identity of these maps in the respective figure legends I can accept this.

Otherwise, the authors have performed a very comprehensive revision and strongly improved the manuscript text (and the figures). All raised issues have been appropriately addressed and resolved. Hence, I recommend publication of the work in *Nature Communications*.

(Remarks on code availability)

Reviewer #3

(Remarks to the Author)

I co-reviewed this manuscript with one of the reviewers who provided the listed reports. This is part of the *Nature Communications* initiative to facilitate training in peer review and to provide appropriate recognition for Early Career Researchers who co-review manuscripts.

(Remarks on code availability)

Reviewer #4

(Remarks to the Author)

The authors have addressed all my concerns. Congratulations to the nice work.

(Remarks on code availability)

Reviewer #5

(Remarks to the Author)

The major strength of this manuscript lies in the cryo-EM data, which include an exceptional electron density map at nearly 2-3 Å resolution of the mature plant mitoribosome, the chloramphenicol-bound ribosome, and the small subunit in complex with the late assembly factor RsgA. The quality and level of detail in presenting new structural features—enabled by the assignment of previously unassigned proteins and their interactions with RNA and other mt-ribosomal proteins—are outstanding and, in my opinion, fully deserve publication in Nature Communications. The structural parallel between the bacterial 30S late assembly factor RsgA and its cauliflower homologue is particularly striking, as are the RNA modification data, which point to the presence of conserved modifications (and potentially conserved modifying enzymes) shared between the plant mitoribosome and its bacterial counterparts.

A notable weakness of the manuscript lies in the presentation of the mt-rRNA modification data. While it is remarkable that the authors found resources to integrate three fundamentally different methodologies to point at presence of modification sites, assign chemical identities, and even tried to distinguish and specify positional isomers of methylated nucleosides (e.g., 2'-O-Me-U, m³U, m⁵U, m³Ψ, as shown in Table 4), the execution raises some concerns. The ambition of this integrative approach is impressive, however, for a number of residues, the modification assignments appear speculative and may rely on assumptions that need to be more clearly articulated and justified.

For example, it is well established that base-calling error analysis from nanopore direct RNA sequencing (DRS) can suggest presence of pseudouridine modifications, but remains largely inconclusive for reliably detecting other types of modifications. If the authors primarily relied on base-call error rates to identify modified positions, it is essential that they clearly define the metric used for such analysis. Based on Supplementary Figure 9a, it appears that a 5–15% error rate difference between native RNA and IVT controls may have been used as a threshold to identify modifications at positions such as G1825, A1827, and A1828. If so, one would expect the number of putative modifications to be significantly higher than those listed in Table 4 and Figure 2. Furthermore, the average error rate across the entire mitochondrial ribosome should be reported to help evaluate the robustness of the method. Another concern is the modest read depth (~2000 reads) for mt-rRNA, which may not provide sufficient statistical power to support low error-rate thresholds used for modification calling.

Furthermore, the authors attempt a comparative nanopore analysis by extracting ion current intensities for IVT and mitochondrial RNA, as shown in SI Figure 10. This approach is indeed considered more sensitive for identifying potential modification sites than simple base-calling error analysis, particularly when statistical tools such as those implemented in xPore or Nanocompore packages are applied. However, the manuscript does not provide any description of the methods, metrics, or statistical thresholds used to analyze the ion current data. I recommend either providing a detailed description or removing the figure.

Nanopore analysis can indicate the presence of a modification at or near a given position—due to the k-mer-based (and not individual base) sensitivity of the ion current signal—but does not reveal the chemical identity of the modification.

Accordingly, Figure 2 Nanopore panel should be revised to reflect only putative modification sites or regions, rather than show definitive assignments. These positions should be presented as candidates for further validation by orthogonal methods : MS and cryo-EM.

Fig 2 captions: “ Colored positions reveal rRNA modification positions detected either by discrepancies between native and IVT sequences or by identification in the cryo-EM structure. Color code corresponds to mis-called bases as shown in Supplementary Fig. 9 and 10” These two statements are in conflict, as color-coding of nanopore coverage data should be done using nanopore data, not cryo-EM identification.

Furthermore, the authors performed LC-MS/MS experiments to investigate approximately 20 modification sites previously suggested by nanopore and cryo-EM analyses. RNase H template directed cleavage was used to target specific regions of interest, meaning that these experiments provide no MS-validation on other regions of the 18S and 26S rRNAs. We call it targeted assessment. With a few exceptions where the signal-to-noise ratio is notably high (e.g., m⁷G525, m⁴Cm1664, U891, m²A2808 spectra with often suboptimal sequence coverage), visual inspection of the MS/MS spectra presented in SI Figure 11 suggests they are of reasonable quality for modification discovery. However, the manuscript lacks any quantitative metric or confidence score to distinguish correct from incorrect sequence and modification position assignments.

Cyanoethylation treatment was employed to differentiate pseudouridine from uridine, and this approach as an only method of pseU mapping may risk significant false-positive identifications. A more critical concern is however that none of the MS/MS spectra presented provide a mean to distinguish positional isomers of methylated nucleotides—for instance, differentiating between ribose methylation (2'-O-Me) and base methylation (e.g., m⁵U, m³U, Um). Therefore, it is problematic that the authors make specific isomeric assignments in Figure 2 (MS panel) and SI Figure 11, which are not directly supported by MS data. Positional isomer assignments have to be done using complete digestion of the gel purified RNase H fragments, followed by nucleoside MS identification.

Finally, the high-resolution cryo-EM data remain the only reliable source for assigning positional isomers of RNA modifications. Although this important point is not sufficiently discussed in the manuscript, but assignment of specific modification chemistries appears to be based not on de novo docking of modifications into high resolution EM densities, but rather on fitting known E. coli modifications (as listed in Table 4) into the cauliflower mitoribosome EM map. This process is accompanied by manual curation (no explicit criteria or validation metrics are provided). That said, I must however

acknowledge that the visualization of 16S and 23S bacterial specific modifications docked so precisely into plant EM densities in Figure 2a is truly impressive. I suggest the authors focus more on this high-confidence structural evidence as a central finding, rather than attempting to map every possible modification site using data of limited resolution or suboptimal quality. Additionally, I strongly encourage the authors to expand the discussion of bacterial rRNA methyltransferase homologs identified in the Brassica oleracea genome (as listed in Table 4), as this could provide valuable biological context and further support for the proposed conservation of modification patterns.

Also, please correct the following minor errors.

Figure 2a modification sites should be labeled as m6,6A or m26A

Table 4. E. coli does not have s2C2501 modification, it is ho5C-2501

Table 4. Please, remove modification chemistry assignments with insufficient evidences (such as Gm-2858, s2C2501). If

Table 4 is the main text table, I would focus solely on consensus modifications, leaving putative sites to SI.

Tables 1,2,3, 5 better belong to SI

SI Figure 11 caption needs revision.

(Remarks on code availability)

We wish to thank the anonymous reviewers for their constructive comments, that have allowed us to improve the content and presentation of our manuscript.

The major changes brought to the manuscript data are as follows:

- (i) All the structural models have been rebuilt with *Brassica oleracea var. botrytis* (true cauliflower) sequences that have become available.
- (ii) An additional analysis of rRNA modifications using mass spectrometry has been performed. This has allowed to resolve most discrepancies between our initial analyses of modifications using cryo-EM and nanopore sequencing.

Furthermore, the main text and the Figures have been extensively edited in accordance with the reviewers' advices, and new Supplementary Figures have been added, to improve the manuscript presentation, i.e. to make it less technical and focus more on the plant-specific features identified here.

A detailed answer to reviewer's comments and a description of changes brought to the manuscript are as follows: Reviewers comments in blue, our answers in black and changes brought to the manuscript text are highlighted in the revised text file.

Reviewer #1

Compared to the structure they published four years ago (Waltz et al., 2020), the authors have now improved the resolution to ~ 2.2 Å. These improved maps allowed the identification of additional ribosomal proteins that were previously unassigned, as well as universal and plant-specific modifications. Furthermore, the authors resolved a chloramphenicol-stalled ribosome with a P-site tRNA and an immature SSU with the assembly factor RsgA.

We thank reviewer 1 for his/her positive appreciation of our work.

1. The assignment of new ribosomal proteins is not particularly interesting, especially since these proteins were already present in the previous low-resolution maps. Are these newly assigned proteins specific to plants? If so, the focus should be placed on those proteins. Otherwise, this section should be significantly reduced.

We still believe that the assignment of new ribosomal proteins is of high interest because the newly assigned proteins here, i.e. mL102, mS77, mS81, mS83 and mL101 all correspond to plant-specific proteins and all belong to the PPR family, a group of eukaryote specific RNA binding proteins, particularly expanded in plants and involved in all steps of mitochondrial gene expression. We had indeed previously identified these proteins in our initial biochemical/proteomic characterization of plant mitoribosome (Waltz et al. 2019), but our previous structural analysis (Waltz et al. 2020) did not allow us to assign their location in the mitoribosome structure. These newly assigned proteins are of particular interest because they shape plant-specific domains of the mitoribosome and one of them, mS77, is predicted to be required for mRNA recruitment. The text has been substantially modified to clarify this point, to focus on the specific functions of these plant-specific PPR proteins and to describe how they contribute to the specialized features defining plant mitoribosomes.

2. The same applies to the chloramphenicol-stalled ribosome. This structure does not provide novel insights into how plant mitoribosome translation occurs. Are there any distinct structural features compared to known mitoribosome structures? If not, this section should also be shortened. If the authors aim to discuss translation states, the research should be focusing on untreated polysomes instead.

Our data do not describe at the structural level the different steps of the translation cycle (initiation, elongation, termination, recycling). Still, our data provides the first snapshot on how mRNA is accommodated in the mRNA channel (especially showing a plant-specific protein, mS86 being involved in this process), how a tRNA is positioned in the decoding center and how a nascent peptide exits the peptide channel. For all these aspects, specificities are observed as compared to bacterial ribosomes or to other mitoribosomes. The text has been substantially modified to focus on the specific features observed here for a plant mitoribosome in translation state.

We wish to point out that, with our mitoribosome system, we cannot focus on "untreated polysomes" to study translation states, in contrast to analysing stalled monosomes as we did. Plant mitoribosomes are membrane bound and soluble polysomes as understood for bacterial or cytosolic ribosomes simply do not occur in

mitochondria. Until now, all the structural data on plant mitoribosomes were obtained for empty mitoribosomes (with no mRNA or tRNA associated). Obtaining structural information on this mitoribosome “in translation state” i.e. in complex with mRNA and tRNA, was only possible after ribosome stalling by chloramphenicol treatment.

Overall, I recommend reorganize these two parts and shifting the focus toward the plant-specific modifications and the RsgA structure.

The main text has been substantially modified to focus more on the plant specific features revealed by the new structures, i.e. RNA modifications and a novel plant specific pathway involving RsgA to prevent mRNA association before the completion of the ribosome small subunit maturation.

3. Although cauliflower and *Arabidopsis* belong to same group, it is very important to build your model basing on the cauliflower genome being used. If the genome is not well annotated, it is necessary to annotate it yourself (collaboration or Biotech company?). Since the authors aim to provide improved models, the precise sequences (proteins and rRNAs) are obligatory. This must be corrected.

This important point has been addressed. While our models from the initial submission were built with *Brassica oleracea* var. *oleracea* (the same species as cauliflower, but another variety) sequences or with sequences of the closely related *Arabidopsis* when *Brassica oleracea* sequences were missing, 2 completely annotated versions of *Brassica oleracea* var. *botrytis* (true cauliflower) have become available during the submission and reviewing process (GWHHDUBS00000000 and GWHBJSH00000000). We were thus able now to use the var. *botrytis* sequences to make models of all the ribosomal proteins, which resulted in few positions to mutate (or even identical proteins).

The ribosomal 5S, 18S and 26S RNA sequences of var. *botrytis*, that we determined experimentally using nanopore technology, had nonetheless already been used in our initial submission model.

The gene IDs of all the r-proteins and rRNAs of *Brassica oleracea* var. *botrytis* are now also provided in Supplementary Tables 2 and 3.

4. It is surprising that the structural and nanopore-identified modification sites are inconsistent. Modifications observed in the structure, especially methylation, should be reliable given the map resolution they achieved. This inconsistency suggests that nanopore sequencing may not be an optimal method for detecting modification sites in this case. The authors should verify these modification sites, particularly the interesting cauliflower-specific ones, using additional classical experimental methods.

The observed inconsistencies between the detection of rRNA modifications by cryo-EM and nanopore sequencing might not be so surprising.

In the case of cryo-EM, although we could get an overall resolution of the mitoribosome at 2.1 Å resolution, this resolution is an average and the local resolution can be higher or lower. At some places, the local resolution was unfortunately not sufficient to assign modifications with certainty (this is now shown in the new Supplementary figure 12). This is particularly true for Ψ that are solely identified by the position of a water molecule, close to a nitrogen atom located at position 3 after uridine isomerisation. For instance, in the 26S Ψ cluster between positions 2229 and 2239, the position of water molecules could not be visualized with certainty.

In the case of nanopore sequencing, modifications are identified through the detection of mismatches (“base calling errors”) between signals obtained with the native sequence and a reference sequence (*in vitro* transcript here). Still, the nanopore does not read sequences base by base, it detects electrical current fluctuations for k-mers, i.e. short series of 5-6 nucleotides. The local sequence context has thus an importance for the detection of modifications. Some sequence contexts associated with some modifications sometimes do not result in a detectable base calling error. This issue has been documented in the literature (e.g. Anreiter et al. (2021) Trends Biotechnol. 39(1):72-89 or Leger et al. (2021) Nat. Commun. 12, 7198).

Nonetheless, in order to try to resolve the inconsistencies between Cryo-EM and nanopore, we have used a mass spectrometry approach to analyze 18S and 26S rRNAs. This has revealed 22 modifications (7 in SSU and 15 in LSU). These new results are shown in a revised Fig. 2c and a new Supplementary Fig. 12. Importantly, these new results have allowed to resolve most discrepancies between the cryo-EM and nanopore results. For instance, MS/MS has confirmed the existence of the Ψ cluster between positions 2188 and 2239 in the LSU.

5. The title needs to be revised. The current title is vague and uninformative. It should be both specific and informative, explicitly mentioning the cauliflower mitoribosome. Meanwhile in the main text, using “flowering plants mitoribosome” is misleading and should be specific.

The title of the manuscript has been revised to: “Structural insights into cauliflower mitoribosome in translation state and in association with a late assembly factor”. We believe that the title is now more specific and more informative. It now explicitly mentions cauliflower.

The term “flowering plant” is a phylogeny term synonymous to “angiosperm”. It describes the major group of land plants that does not include more basal plants such as gymnosperms, ferns or mosses. “Flowering plant” might nonetheless be misleading and has been replaced throughout either by “cauliflower” or by “angiosperm” to be more specific.

6. Does RsgA exist in fungi or animal mitochondria, or is it specific to plants? Clarifying this point would add value to the manuscript. Additionally, comparing the RsgA structure with the known bacterial structure would provide insight into what is new and important in this study.

While RsgA is widely distributed in bacteria, it is not well conserved in eukaryotes. In particular, it is not present in animals and yeast while it is present in plants. Precisions on this distribution have been added in the main text. As previously described in the text, it is precisely because plant RsgA has a specific C-terminal extension, not present in bacteria, that it is able to block the mRNA channel before completion of the SSU maturation. This has been further emphasized in the text and a new figure, Supplementary figure 13 compares RsgA across species, to highlight the plant-specific additional domain. This point is of particular importance, as it identifies a novel plant-specific process to prevent mRNA association before completing SSU maturation.

In *Arabidopsis thaliana*, there are two isoforms of RsgA: RsgA1 and RsgA2. Notably, only RsgA1, which is used in the model, contains an extra C-terminal extension. What is the situation in cauliflower? Once again, please build the cauliflower model.

Both RsgA1 and 2 have the C-terminal extension in *Arabidopsis* (see the figure below). In cauliflower (*Brassica oleracea* var. *botrytis*) a single isoform of RsgA is found. It is nearly identical to both *Arabidopsis* RsgA isoforms, with the additional C-terminal extension.

Here are some suggestions to improve the manuscript:

1. The abstract needs significant revision. For instance, the sentence “However, the molecular mechanisms of translation by plant mitoribosomes remain largely unknown” does not align with the content of this manuscript, as this problem is not addressed here. Additionally, the use of “extensive” to describe the modifications is misleading, as only dozens of modified sites were identified. Please remove or revise this phrasing in the abstract and throughout the manuscript.

The abstract has been modified accordingly, to reflect better the content of the manuscript. The term “extensive” has been removed or replaced by precise numbers throughout.

2. In line 82, specify which sample and dataset the described structures are derived from.

This has been clarified in the text.

3. In Figure 2a, please show all (also could be in supplementary figures) the identified modifications supported by density maps to provide comprehensive visualization.

All densities corresponding to rRNA modifications are now shown in a revised Figure 2a.

4. In Figure 3b, the nascent chain appears suspicious as it seems disconnected from the CCA end. In the human mitoribosome, a mitoribosome protein binds in the peptide exit tunnel and reaches the PTC. Please check if such a protein is present in cauliflower mitoribosome

In human, the tail of mL45 indeed localizes in the peptide path in the absence of OXA1 interaction, thus preventing peptide exit when mitoribosomes are not tethered to the membrane (Itoh et al. 2021). This process does not occur in plants because mL45 is not present in the plant mitoribosome.

5. Figure 3e, the inset is not prominent and difficult to catch. Consider adding a frame around the inset to improve clarity. In general, please avoid using white shading to separate panels.

We revised figure 3 to remove the peptide path and focus on a comparison between the structure with and without mRNA (originally shown as a small inset in figure 3e).

6. Figures S5 and S6, the overall views do not help. Use consistent colors for all proteins and label them properly for better comprehension.

Figures S5 and S6 have been revised accordingly.

7. For the data processing section, provide the model-to-map FSC and include model resolution statistics in Table 1. Additionally, for Table 1, remove statistics for focus-refined maps as they do not add value and make the table harder to read.

Table 1 has been revised accordingly.

8. The interaction between m2A2808 and A2377 mentioned in line 206, as well as the two PPR modules described in line 132, should be illustrated with additional figures.

The 2 PPR modules of mS81 are shown in orange in Figure 1d. This has been clarified in the text. The text has been shortened and the mention of the interaction between m2A2808 and A2377 has been removed (the feature is present but not discussed)

9. The methodology for GTPγS treatment is unclear, especially since it is hard to believe it was done in the same way as chloramphenicol treatment. Furthermore, the chloramphenicol concentration is inconsistent—500 μg/mL or 200 μg/mL?

The chloramphenicol treatment was 200 μg/mL, while the GTP analog treatment was 200 μM This has been corrected in the text.

Reviewer #2

I think that the work significantly expands our knowledge of plant mitoribosomes, and how they compare to other (mito)ribosomes. Therefore, I find reviewed manuscript suitable for the publication in Nature Communications, provided that the authors resolve the two main issue and consider addressing the listed minor issues. Of note, I do not question the scientific validity of the presented results but would like the authors to improve the clarity of the manuscript for a broad audience.

We thank reviewer 2 for his/her positive appreciation of our work.

Major issues

1. To improve the map quality in regions of particular interest and of newly identified components, the authors performed several local refinement steps. Subsequently, they used these locally refined maps to generate composite maps – such types of maps are sometimes called “Frankenstein maps”. The usefulness of such local refinement steps is undisputed, but the generation of composite maps is heavily debated in the field. I don’t see any advantage of using such artificially stitched maps over globally refined maps. Showing locally refined maps can of course be used to zoom into the locally refined area. I understand the temptation to generate prettier maps for prettier figures but combining several locally refined maps into a composite map (e.g. Fig. 1a/b, Fig3a and Fig.4a) does not relate directly to the obtained data. I would encourage the authors to replace those maps with the obtained global maps. At the obtained resolution, the maps will still look nice, but it will be become more obvious for the non-experts, where the author identified dynamic movements.

Concerning the composite maps, their use has indeed been actively debated in the field until 2020 (<https://doi.org/10.48550/arXiv.2311.17640>). It is now entirely accepted and even recommended for entries depositions to the Electron Microscopy Data Bank (EMDB). The guidelines proposed for EMDB composite maps entries (https://www.ebi.ac.uk/emdb/documentation/deposition/composite_map) have been thoroughly applied here. Nonetheless, the global maps were already also included in the manuscript in Supplementary Figures 1 to 4. It is noteworthy that the global maps simply do not allow to visualize all the mitoribosome features, i.e. the highly mobile head extension domain of the SSU.

2. I don't fully understand why the authors would not show the densities for all modified rRNA nucleotides.

The densities for all modified rRNA nucleotides are now indicated in a revised Figure 2.

They should also explain to the reader, how they are able to discriminate between U and Pseudo-U.

In the cryoEM structures, Ψ are solely identified by the position of a water molecule, close to a nitrogen atom located at position 3 of the base after uridine isomerisation. This is now explained in the text.

I would suggest expanding Figure 2a by showing the densities for all modified nucleotides in the LSU.

The densities for all modified rRNA nucleotides are now indicated in a revised Figure 2.

In addition, I would encourage the authors to present the densities for all positions that were detected by nanopore sequencing, which could not be confirmed by cryo-EM – I guess, this could be shown a Suppl. Figure.

The densities corresponding to all positions that were detected by nanopore sequencing are now presented in Supplementary Figure 12. The local resolution of the structure is however insufficient at these site to confirm nanopore results.

I am particularly curious about the density around SSU position 959. This would allow the reader to understand the rational behind the decision of the authors to call or not to call a modification at a certain position.

When using nanopore sequencing, modifications are identified through the detection of mismatches (“base calling errors”) between signals obtained with the native RNA sequence and a reference sequence (*in vitro* transcript here). For instance, nanopore shows very clearly that while an adenosine is present at position 959 of the 18S rRNA in the *in vitro* transcript, in accordance with the genomic sequence, this residue is not a canonical A in the native rRNA, thus revealing an RNA modification. However, for 959 (and 961 which could also be modified based on homology with bacterial ribosomes), inspection of the density (shown in Supplementary Figure 12b) does not reveal any obvious modification. Still while for some modifications, e.g. Ψ where there is abundant literature describing that Ψ are mostly misread as C (e.g. Fleming ACS Chem Biol 2023), for other modifications such as the one at position 959, nanopore results alone do not allow to identify the precise nature of the modification.

I would also like to hear the opinion of the authors for those modifications that can be detected by cryo-EM, but not by Nanopore sequencing. At the moment, the whole RNA-modification field seems to be excited about Nanopore sequencing, but apparently some highly modified sites are not detected – this kind of information about the weakness of nanopore-based detection should reach the interested expert reader.

As indicated in the response to reviewer 1 comments, the nanopore does not read sequences base by base, it detects electrical current fluctuations for k-mers, i.e. short series of 5-6 nucleotides. The local sequence context has thus an importance for the detection of modifications. Some sequence contexts associated with some modifications sometimes do not result in a detectable base calling error. This issue has been documented in the literature (e.g. Anreiter et al. (2021) Trends Biotechnol. 39(1):72-89 or Leger et al. (2021) Nat. Commun. 12, 7198). This is now discussed in the main text.

Nonetheless, in order to try to resolve the inconsistencies between Cryo-EM and nanopore, we have used a mass spectrometry approach to analyze 18S and 26S rRNAs. This has revealed 20 rRNA modifications, 7 in SSU and 13 in LSU. These new results are shown in a revised Fig. 2c and a new Supplementary Fig. 11. Importantly, these new results have allowed us to resolve most discrepancies between the cryo-EM and nanopore results. For instance, MS/MS has confirmed the existence of the Ψ cluster between positions 2188 and 2239 in the LSU.

Minor issues

- Typically, ribosomes are purified either from the monosome or polysome fractions. For people from outside the ribosome field an additional supplementary panel or figure explaining the sample preparation process would help.

Mitoribosomes studied here are membrane bound ribosomes. Polysomes as understood for bacterial or cytosolic ribosomes simply do not occur in mitochondria. The purification of ribosome fractions from isolated mitochondria results in 2 peaks only, one corresponding to monosomes and one corresponding to dissociated SSU and LSU. This is shown in details in our previous publication (Waltz et al. Nat Plants 2019).

- In Supplementary Table 1 there is no magnification value specified for the 9EVS structure – please complete. Actually, it states “???” , which does not look very scientific.

Table 1 has been revised accordingly.

- Fig. 1a and 1b – both panels are not referenced in the main text of the manuscript.

Fig. 1a, Fig. 1b are now cited in the main text.

- Fig. 1c - please provide labels for each of the visible ions.

The nature of ions shown as red or green dots in Fig. 1c is now explained in the Figure caption.

- Fig. 1e panel is introduced in the text before Fig. 1d.

Fig. 1d and Fig. 1e have been exchanged correctly re-labelled in the text and figure caption.

- Fig. 2c - please consider changing the grey color of labels above grey background.

We think that this color code (grey on light grey and black on dark grey) helps to easily visualize which modification was detected by which method.

- Fig. 2c legend – missing space “.. grey.”?” “..”.

This has been corrected.

- Fig. 3a and 3c panels are not referenced in the main text of the manuscript.

Fig. 3a and 3c are now referenced in the main text.

- Fig. 3d panel is referenced in the main text before Fig. 3b.

Fig. 3b is now cited before Fig. 3d in the main text.

- Labels SSU/LSU in Fig. 3a have different colors than the same labels in Fig. 2. Please consider improving the consistency between main and supplementary figures i.e. same colors for mitoribosome components in all figures.

Labels have now consistent colors throughout the Figures.

- Fig. 3b - Potassium ions should be labeled.

Potassium ions are now described in the Figure caption.

- Page 6 - Consider rephrasing “peptide exit via the peptide tunnel” to “...the end of exit tunnel”

This has been rephrased accordingly.

- The label “d” in Fig. 4a should be highlighted or removed.

We believe that this label is visible enough and has to be included.

- Fig. 4b panel is not referenced in the main text of the manuscript.

Fig. 4b is now referenced in the text.

- The panel referencing order of Fig. 4 in the main text is a, g, d, e and c.

This has been fixed.

- For Supplementary Fig. 1-4 please consider pipelines simplification, specify types of particle picking, remove redundant 2D averages, show scales of 2D averages, for each density include orientational bias, show all used classification masks, label each local refinement mask

We kept the pipeline as they are, but added orientation plots for the main maps, and labelled all refinement masks.

- Use 2 colors for the gradient in the local resolution estimations – here white is higher res than red, which might be misleading to the non-experts. Local resolutions should be calculated for the same FSC cutoff as reported GSFSCs (0.143).

We revised the local resolution plots using a different color gradient (both color blind friendly and without white) and used the 0.143 cutoff.

- Some weird box-artefact in Suppl. Fig. 3b can be observed

We do not see the box artefacts in questions.

- Suppl. Fig. 5a - bL25-2m has different color than the same protein in Fig. 1e.

Colors are now consistent between these figures (and we tried to make color as consistent as possible between figures in general).

- Suppl. Fig. 5c – maybe H4 should also be in bold.

H4 is now shown in bold.

- Labels for p56 and p58 are missing in Suppl Fig. 5d.

Labels for p56 and p58 have been added in Suppl Fig. 5d

- In Supplementary Fig. 6a legend “.. of domain I, forming p17, p17 and p21-22, specific ..” should rather be “.. of domain I, forming p15, p17 and p21-22, specific ..” ...but maybe I misunderstood.

One of the p17 has been corrected to p15.

- Suppl. Fig. 7c – label for mRNA exit channel can hardly be read – consider reformatting.

The label for the mRNA exit channel is now shown in a more visible color.

- Did the authors analyzed also the large subunit fraction isolated from the mitoribosomes purified in the presence of non-hydrolysable GTP analogue to see whether any additional factors are binding?

No additional factor could be observed in the other fractions from the sample treated with the GTP analog.

- Supplementary Table 1 is not referenced in the main text of the manuscript.

Table 1 is now referenced in the main text.

- In Supplementary Table 1 local resolutions should be specified for FSC=0.143, as indicated for global resolution.

Table 1 has been modified accordingly.

- Supplementary Table 4 should also be referenced in the main text of the manuscript, not only in the legend of Fig. 2.

Table 4 is now also referenced in the main text.

- Supplementary Fig. 6c panel is referenced in the main text before Supplementary Fig. 6a and 6b.

This has been fixed.

- Supplementary Fig. 8b panel is referenced in the main text before Supplementary Fig. 8a.

This has been fixed.

- Supplementary Fig. 9 and 10 should also be referenced in the main text of the manuscript, not only in the Fig. 2 legend.

Supplementary Fig. 9 and 10 are now also referenced in the main text.

Of note – during my search, I got confused whether the previous PDB entry 6XYW shows the cauliflower or *A. thaliana* mitoribosome. The paper clearly relates to cauliflower, but the PDB entry list *A. thaliana* as the organism...could the authors resolve this confusing issue with the PDB and EMDB?

Thank you for pointing out this problem. The PDB entry 6XYW does correspond to a cauliflower mitoribosome structure, and not *Arabidopsis*. We contacted the PDB to have this corrected.

Reviewer #3

We thank reviewer 3 for his/her participation to the reviewing process that allowed us to improve our manuscript.

Reviewer #4

In the study, high resolution Cryo-EM structural data were collected for the mitochondrial ribosome of cauliflower. The data were collected from ribosomal particles in three different states, including ribosomes with bound tRNA at the P-site and intermediate assembly states. Due to the high quality of the data, the authors were able to identify new sections, which were previously not resolved, e.g. densities of additional ribosomal proteins. Interestingly, rRNA modifications could be identified, which were previously not described for mitochondrial ribosomes.

We thank reviewer 4 for his/her positive appreciation of our work.

Previously, several structural data of mitochondrial ribosomes from algae, Arabidopsis and cauliflower contributed to a good and high-resolution understanding about mitochondrial ribosomes. This also includes work from the same group. Certainly, the findings about the rRNA modifications and the assembly intermediate states are interesting. However, it needs to be better elaborated how the discovered features really contribute to a deeper and broader understanding about ribosome biology in plant mitochondria.

The manuscript text has been extensively modified, as described below, to focus much more on the conceptual advances brought by the work that we present here.

The text reads very technical and needs a broader perspective. By reading the current version, several questions arise:

- Which of the unknown aspects are now solved? The way it is written, it appears that the findings are just additions to the known features (although the authors tried to mention this but this reads very vague).

We have substantially edited the text to make it sound less technical and to focus on the answers to biological questions that the data presented here has brought.

In summary: The conceptual advances brought by this manuscript are three-fold :

(i) It uncovers the full set of rRNA modifications in plant mitoribosomes, revealing 3 to 6 times more modifications as compared to yeast or mammal mitoribosomes. Some of them are completely specific to plants. The functional consequence of their occurrence is now discussed.

(ii) It identifies a novel plant specific process, involving RsgA, to prevent mRNA association before the completion of the ribosome small subunit maturation.

(iii) It reveals how translation proceeds, with the positioning of the mRNA in its channel and its interaction with a tRNA in the mitoribosome P-site. Here as well, some observed features are specific to plants and differ from bacterial ribosomes and from mitoribosomes from other eukaryotes.

- Are these new findings specific to cauliflower mito ribosomes? Which features are common for plant cells and how do the data help to generally understand specific features of (plant) mito ribosomes?

It can be anticipated that the new finding described here for cauliflower (structures of plant specific PPR proteins, specificities of the mRNA channel and tRNA binding, new process involving RsgA) can be generalized to the entire group of angiosperms / flowering plants, i.e. the vast family of land plants that do not include more basal plants such as gymnosperms, fern or mosses, or at least to dicotyledon plants. Our previous phylogeny analyses (Waltz et al. Mitochondrion 2020) had already shown that the full set of specific PPR proteins occurring in cauliflower is conserved in angiosperms. Likewise, we now provide an additional phylogeny analysis of RsgA to show that the C-term extension specific to plants as compared to bacteria, associated to the mitochondrial localization of RsgA, is specific of dicotyledon plants. This is shown as a new supplementary Fig. 13 and described in the text.

- How do these ribosomes functionally differ from 80S ribosomes, bacterial or plastidic 70S ribosomes. In the results and discussion chapter, the authors only briefly tried to compare their findings to other ribosomal structures. I would support publication in the journal if these points can be clarified.

More discussion has been added to describe how the plant mitoribosome functionally differ from cytosolic ribosomes, bacterial ribosomes and plastid ribosomes.

Reviewer #1

We highly appreciate that the authors have rebuilt their structural models based on the annotated genome of *Brassica oleracea* var. botrytis and complemented their findings with mass spectrometry to confirm rRNA modification sites. The manuscript has also been substantially rewritten to highlight plant-specific features.

We thank reviewer 1 for his/her positive appreciation of our work.

1. In the section “Cauliflower mitoribosome contains a set of at least 20 rRNA modifications”, the phrase “at least” remains ambiguous. We expect the authors to clearly define which modifications are robustly supported, so that readers can confidently interpret the data. It is not sufficient to present all detected sites and leave it to the reader to determine which are reliable.

We have edited the manuscript to further clarify that the 19 consensus modifications, identified by at least two methods are the ones that are robustly supported and that we consider non ambiguous.

Examples are 891, 1105, 2762, 2803, they are shown in the main figure, should we trust them?

Modifications at positions such as 891 and 1105, although clearly seen in the cryo-EM data could not be confirmed by another method and as such are not considered to be robustly supported

As a suggestion, in Supplementary Figure 11, please label which peaks correspond to modified signals in the MS data.

Supplementary Figure 11 has been modified. Peaks corresponding to modified signals are now indicated by red stars in MS/MS spectra.

2. In the section “Identification of plant-specific features of the translating mitoribosome”, although the authors have addressed the reviewer’s points in their rebuttal, the plant-specific features are still not easily to get for readers who are not experts in the mitoribosome field. To improve clarity, we suggest adding structural comparisons or schematic overlays with other mitoribosomes to clearly highlight the unique plant-specific features.

We believe that the specificities of the plant mitoribosome as compared to other eukaryote mitoribosomes have already been described in details in different reviews, by our group and others e.g. Scaltsoyiannes et al. 2022 doi: 10.3390/ijms23073474.

Still, in order to clarify this point, we have changed the text to highlight the fact that all the rPPR proteins occurring in this mitoribosome are specific to plants. We have also modified Table 3 to indicate all the plant specific proteins.

3. The UniProt ID F4HTL8 corresponds to RsgA2 in *Arabidopsis thaliana*, which lacks the C-terminal extension. It remains unclear whether homologs in *Brassica oleracea* all possess such an extension.

UniProt ID F4HTL8 is annotated as RsgA2 due to its homology with RsgA, mainly its G-domain. RsgA2 not only lacks the C-terminal extension of RsgA but also the Zn-domain that mediates the interaction with the SSU head (López-Alonso et al., 2017; Razi et al., 2017). Thus, it is yet to be determined whether F4HTL8 is a true homolog of RsgA and whether it interacts with the *Arabidopsis* mitoribosome. Nevertheless, all homologs in *Brassica oleracea* possess the Zn-domain and the specific C-terminal extension.

4. There is inconsistency in the naming of the bL25m isoform, referred to as “bL25m-2” in Line 115 but as “bL25-2m” in Figure 1 and Supplementary Figure 5b.

The text has been corrected for consistency with the Figures.

Reviewer #2

... the authors have performed a very comprehensive revision and strongly improved the manuscript text (and the figures). All raised issues have been appropriately addressed and resolved. Hence, I recommend publication of the work in Nature Communications.

We thank reviewer 2 for his/her positive appreciation of our work.

Reviewer #3

I co-reviewed this manuscript with one of the reviewers who provided the listed reports...

We thank reviewer 3 for his/her participation in the reviewing process.

Reviewer #4

The authors have addressed all my concerns. Congratulations to the nice work.

We thank reviewer 4 for his/her positive appreciation of our work.

Reviewer #5

... The quality and level of detail in presenting new structural features—enabled by the assignment of previously unassigned proteins and their interactions with RNA and other mt-ribosomal proteins—are outstanding and, in my opinion, fully deserve publication in Nature Communications. ...

We thank reviewer 5 for his/her positive appreciation of our work.

A notable weakness of the manuscript lies in the presentation of the mt-rRNA modification data. While it is remarkable that the authors found resources to integrate three fundamentally different methodologies to point at presence of modification sites, assign chemical identities, and even tried to distinguish and specify positional isomers of methylated nucleosides (e.g., 2'-O-Me-U, m³U, m⁵U, m³Ψ, as shown in Table 4), the execution raises some concerns. The ambition of this integrative approach is impressive, however, for a number of residues, the modification assignments appear speculative and may rely on assumptions that need to be more clearly articulated and justified.

The text has been modified to clearly state that the assignment is based on conserved *E. coli* modifications when other data are missing. Line 194: “The aforementioned approaches were employed to detect the position and the type of each modification. When the latter was not adequately supported by our experimental methods, the type of modification was assigned according to its equivalent residue of the *E. coli* ribosome, for the conserved modifications (Supplementary Table 4), or left unassigned for the plant-specific ones.”

For example, it is well established that base-calling error analysis from nanopore direct RNA sequencing (DRS) can suggest presence of pseudouridine modifications, but remains largely inconclusive for reliably detecting other types of modifications. If the authors primarily relied on base-call error rates to identify modified positions, it is essential that they clearly define the metric used for such analysis.

We agree with reviewer 5. The base-call error (for cases other than Ψ) is primarily used for the identification of the modified position and not the type of the modification.

Based on Supplementary Figure 9a, it appears that a 5–15% error rate difference between native RNA and IVT controls may have been used as a threshold to identify modifications at positions such as G1825, A1827, and A1828. If so, one would expect the number of putative modifications to be significantly higher than those listed in Table 4 and Figure 2.

The text in methods has been modified to clarify that “The difference in base-calling error between native and IVT transcripts ($\Delta\text{CallError}$, %) was then computed to identify positions with putative modifications. Positions with a $\Delta\text{CallError}$ greater than 10% were considered indicative of RNA modifications. This 10% threshold corresponds to approximately three times the average background miscalling level.”

Positions G1825 and A1828 were removed from the list of Nanopore-identified positions as they presented $\Delta\text{CallError}$ below 10%.

Additionally, Supplementary Table 7 listing the $\Delta\text{CallError}$ for all positions of interest is now provided.

Furthermore, the average error rate across the entire mitochondrial ribosome should be reported to help evaluate the robustness of the method.

The average base calling error across the entire *Brassica oleracea* mitoribosome, as quantified by non-reference base detection, was 2.9% for in vitro transcript rRNA sequencing (i.e., 26S IVT rRNA, 2.91%; 18S rRNA IVT, 2.92%)

and 3.5% for Native rRNA sequencing (i.e., 26S Native rRNA 3.66%; 18S Native rRNA, 3.25%). These values exclude insertion-deletion type errors, which were not considered for identification of rRNA sites of interest.

Another concern is the modest read depth (~2000 reads) for mt-RNA, which may not provide sufficient statistical power to support low error-rate thresholds used for modification calling.

On the contrary, 2000x read depth is a very robust dataset for identifying miscalled bases as sites of interest in ONT direct RNA sequencing. For example, Tavakoli et al. (2023) Nat Commun showed Guppy basecaller U-to-C miscall error (our approach) works as a proxy for U-modifications and confidently detected pseudouridylation (Ψ) with >400 reads (referred to as “high coverage”). Moreover, Leger et al. (2021) Nat Commun. reproducibly detected modified RNA bases with 500-2000 reads in benchmarking. Finally, Delgado-Tejedor et al. (2024) Nat Commun. detected specific rRNA modifications using coverages of 50-500 reads, depending on the pipeline. Furthermore, the authors attempt a comparative nanopore analysis by extracting ion current intensities for IVT and mitochondrial RNA, as shown in SI Figure 10. This approach is indeed considered more sensitive for identifying potential modification sites than simple base-calling error analysis, particularly when statistical tools such as those implemented in xPore or Nanocompore packages are applied. However, the manuscript does not provide any description of the methods, metrics, or statistical thresholds used to analyze the ion current data. I recommend either providing a detailed description or removing the figure.

Such packages were not used here. The ion current intensities are presented for each position, but they were not utilized further for the assignment of the modifications.

Nanopore analysis can indicate the presence of a modification at or near a given position—due to the k-mer-based (and not individual base) sensitivity of the ion current signal—but does not reveal the chemical identity of the modification. Accordingly, Figure 2 Nanopore panel should be revised to reflect only putative modification sites or regions, rather than show definitive assignments. These positions should be presented as candidates for further validation by orthogonal methods : MS and cryo-EM.

We agree that DRS does not reveal the chemical identity of the modification. As mentioned above, assignment was performed according to conserved modifications of *E. coli* when relevant. This is now clearly indicated in the text.

Fig 2 captions: “Colored positions reveal rRNA modification positions detected either by discrepancies between native and IVT sequences or by identification in the cryo-EM structure. Color code corresponds to mis-called bases as shown in Supplementary Fig. 9 and 10” These two statements are in conflict, as color-coding of nanopore coverage data should be done using nanopore data, not cryo-EM identification.

Figure 2 caption has now been corrected to “All positions of interest—i.e. indicated as modified by any of the detection methods—are presented as colored percentage stacked bars in the otherwise grey read depth chart. Color categories in such highlighted bars correspond to four possible base calls obtained at the rRNA position: Green, A; Red, U; Orange, G; and/or Blue, C. Non-reference base calls are positions of interest identified by nanopore direct RNA sequencing that may indicate base modifications.”

Furthermore, the authors performed LC-MS/MS experiments to investigate approximately 20 modification sites previously suggested by nanopore and cryo-EM analyses. RNase H template directed cleavage was used to target specific regions of interest, meaning that these experiments provide no MS-validation on other regions of the 18S and 26S rRNAs. We call it targeted assessment. With a few exceptions where the signal-to-noise ratio is notably high (e.g., m⁷G525, m⁴Cm1664, U891, m²A2808 spectra with often suboptimal sequence coverage), visual inspection of the MS/MS spectra presented in SI Figure 11 suggests they are of reasonable quality for modification discovery. However, the manuscript lacks any quantitative metric or confidence score to distinguish correct from incorrect sequence and modification position assignments.

No metric or confidence score was used to assign modified positions by MS. As reviewer 5 mentions, template directed cleavage was indeed used to verify the occurrence of rRNA modifications at places suggested by cryo-EM and / or DRS data.

Cyanoethylation treatment was employed to differentiate pseudouridine from uridine, and this approach as an only method of pseU mapping may risk significant false-positive identifications. A more critical concern is however that none of the MS/MS spectra presented provide a mean to distinguish positional isomers of methylated nucleotides—for instance, differentiating between ribose methylation (2'-O-Me) and base methylation (e.g., m⁵U, m³U, Um). Therefore, it is problematic that the authors make specific isomeric assignments in Figure 2 (MS panel) and SI Figure 11, which are not directly supported by MS data. Positional isomer assignments have to be done using complete digestion of the gel purified RNase H fragments, followed by nucleoside MS identification.

We fully agree with reviewer 5, that MS does not allow to distinguish positional isomers of methylated nucleotides. Isomeric assignments have been included in the MS panel of Figure 2 according to equivalent bacterial modifications. This has been clarified in the caption of Figure 2.

Finally, the high-resolution cryo-EM data remain the only reliable source for assigning positional isomers of RNA modifications. Although this important point is not sufficiently discussed in the manuscript, but assignment of specific modification chemistries appears to be based not on de novo docking of modifications into high resolution EM densities, but rather on fitting known *E. coli* modifications (as listed in Table 4) into the cauliflower mitoribosome EM map. This process is accompanied by manual curation (no explicit criteria or validation metrics are provided). That said, I must however acknowledge that the visualization of 16S and 23S bacterial specific modifications docked so precisely into plant EM densities in Figure 2a is truly impressive.

We thank reviewer 5 for his positive appreciation.

Additionally, I strongly encourage the authors to expand the discussion of bacterial rRNA methyltransferase homologs identified in the *Brassica oleracea* genome (as listed in Table 4), as this could provide valuable biological context and further support for the proposed conservation of modification patterns.

The discussion of bacterial rRNA methyltransferase homologs has been expanded as follows. “This also suggests that many bacterial modification enzymes must be conserved in plants (putative enzymes based on homology are presented in Supplementary Table 4), or have been replaced by new ones, to maintain these modifications.”

Also, please correct the following minor errors.

Figure 2a modification sites should be labeled as m⁶,6A or m²6A

Figure 2a has been corrected with m⁶₂A

Table 4. *E. coli* does not have s²C2501 modification, it is ho⁵C-2501

s²C2501 has been corrected to ho⁵C2501.

Table 4. Please, remove modification chemistry assignments with insufficient evidences (such as Gm-2858, s²C2501). If Table 4 is the main text table, I would focus solely on consensus modifications, leaving putative sites to SI.

s²C2501 has been corrected to ho⁵C2501. A note was added in Table 4 for Gm2858: “The type of modification was assigned according to the human mitoribosome”.

Tables 1,2,3, 5 better belong to SI

All tables are already in SI.

SI Figure 11 caption needs revision.

Supplementary Figure 11 caption has been revised